# Uncovering motifs of concurrent signaling across multiple neuronal populations

**Evren Gokcen**[1], **Anna I. Jasper**[2], **Alison Xu**[2],
**Adam Kohn**[*,2,3,4], **Christian K. Machens**[*,5], **Byron M. Yu**[*,1,6]
[1]Dept. of Electrical and Computer Engineering, Carnegie Mellon University
[2]Dominick Purpura Dept. of Neuroscience, [3]Dept. of Ophthalmology and Visual Sciences,
[4]Dept. of Systems and Computational Biology, Albert Einstein College of Medicine
[5]Champalimaud Neuroscience Programme, Champalimaud Foundation
[6]Dept. of Biomedical Engineering, Carnegie Mellon University
`egokcen@cmu.edu`, `{anna.jasper, alison.xu, adam.kohn}@einsteinmed.edu`,
`christian.machens@neuro.fchampalimaud.org`, `byronyu@cmu.edu`
[*]Denotes equal contribution.

## Abstract

Modern recording techniques now allow us to record from distinct neuronal populations in different brain networks. However, especially as we consider multiple (more than two) populations, new conceptual and statistical frameworks are needed to characterize the multi-dimensional, concurrent flow of signals among these populations. Here, we develop a dimensionality reduction framework that determines (1) the subset of populations described by each latent dimension, (2) the direction of signal flow among those populations, and (3) how those signals evolve over time within and across experimental trials. We illustrate these features in simulation, and further validate the method by applying it to previously studied recordings from neuronal populations in macaque visual areas V1 and V2. Then we study interactions across select laminar compartments of areas V1, V2, and V3d, recorded simultaneously with multiple Neuropixels probes. Our approach uncovered signatures of selective communication across these three areas that related to their retinotopic alignment. This work advances the study of concurrent signaling across multiple neuronal populations.

## 1 Introduction

Cortical circuits functionally involve feedforward, feedback, and horizontal interactions between many neuronal populations that span distinct areas and layers. Recording techniques now allow us to record from many neurons across these populations [1–3] (Fig. 1a). To capitalize on the scientific opportunities presented by these recordings, however, new conceptual and statistical frameworks are needed, particularly as we consider communication across multiple (more than two) populations.

Characterizing interactions between just two populations is a challenging high-dimensional problem. Dimensionality reduction techniques have therefore been increasingly used for this purpose [4–7]. Methods like canonical correlation analysis (CCA) [8] and its probabilistic variants [9], in particular, identify a low-dimensional set of latent variables that parsimoniously describe the interactions between two populations. This cross-population shared-latent model has inspired several extensions targeted toward neural recordings [10–14].

Communication between two populations, however, occurs bidirectionally and likely concurrently, and disentangling this concurrent communication is a substantial challenge in neuroscience. A

37th Conference on Neural Information Processing Systems (NeurIPS 2023).

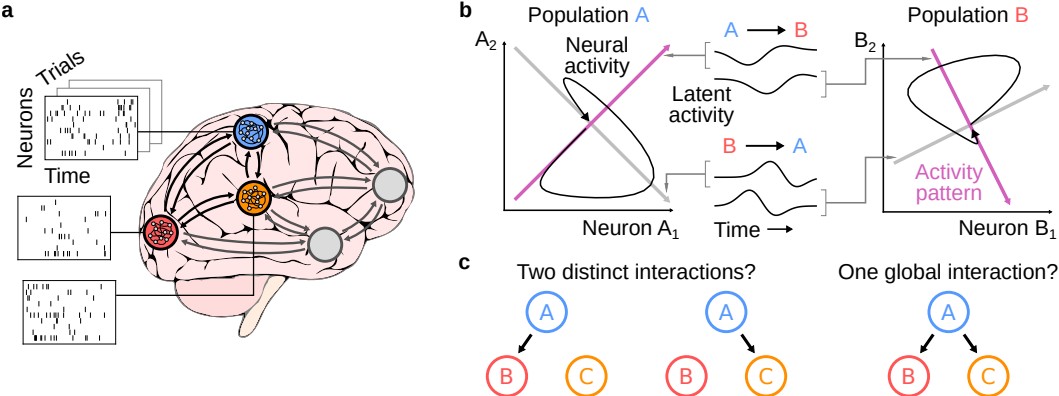

Figure 1: Challenges in studying concurrent signaling across multiple populations of neurons. (**a**) High-dimensional, simultaneous recordings across multiple neuronal populations. (**b**) Disentangling concurrent signal flow. (**c**) Distinguishing network-level interactions.

recently developed dimensionality reduction approach, delayed latents across groups (DLAG) [15], addresses this challenge by leveraging two insights: (1) communication between two populations is not instantaneous, and (2) while concurrently relayed signals might be difficult to tease apart from the raw neural activity (Fig. 1b, black trajectories in each population space), they could be pinpointed along certain dimensions or axes (Fig. 1b, "latent activity" measured along the magenta and gray dimensions; note how activity in one population leads activity in the other population).

Pairwise methods are ill-equipped, however, to analyze communication across three or more recorded populations. Suppose, for example, that we wish to study the interactions of three recorded populations, A, B, and C (Fig. 1c). One could consider applying one of the aforementioned methods to each pair of populations. However, we would encounter the following interpretational ambiguity. Suppose that populations A and B exhibit shared activity fluctuations, and populations A and C also exhibit shared fluctuations. Do populations A, B, and C all co-fluctuate together (Fig. 1c, right; black arrows: direction of influence)? Or do A and B co-fluctuate in a way that is uncorrelated with the way in which A and C co-fluctuate (Fig. 1c, left)? Only by analyzing all populations together can we differentiate these possibilities. Multi-population dimensionality reduction approaches, such as group factor analysis (GFA) [16], could be applied toward that end, but the challenge of disentangling concurrent, bidirectional signaling remains. New approaches are needed to jointly characterize the multi-dimensional, concurrent flow of signals among multiple populations.

We therefore propose multi-population DLAG (mDLAG), a dimensionality reduction framework that determines (1) the subset of populations described by each latent dimension, (2) the direction of signal flow among those populations, and (3) how those signals evolve over time within and across experimental trials. We illustrate these features in simulation, and further validate mDLAG by applying it to previously studied recordings from neuronal populations in macaque visual areas V1 and V2. Then we study interactions across select laminar compartments of areas V1, V2, and V3d, recorded simultaneously with multiple Neuropixels probes. mDLAG uncovered signatures of selective communication across these three areas that related to their retinotopic alignment. Throughout the analyses of simulated data and neural recordings, mDLAG provided improved interpretation and quantitative performance relative to alternative approaches. This work advances the study of concurrent signaling across multiple neuronal populations and its role in brain function.

## 2 Delayed latents across multiple groups (mDLAG)

**Observation model and automatic relevance determination** For population $m$ (comprising $q_m$ neurons) at time $t$ on trial $n$, we define a linear relationship between observed activity, $\mathbf{y}_{n,t}^m \in \mathbb{R}^{q_m}$, and latent variables, $\mathbf{x}_{n,t}^m \in \mathbb{R}^p$ (Fig. 2a):

$$\mathbf{y}_{n,t}^m = C^m \mathbf{x}_{n,t}^m + \mathbf{d}^m + \boldsymbol{\varepsilon}^m \tag{1}$$

$$\boldsymbol{\varepsilon}^m \sim \mathcal{N}(\mathbf{0}, (\Phi^m)^{-1}) \tag{2}$$

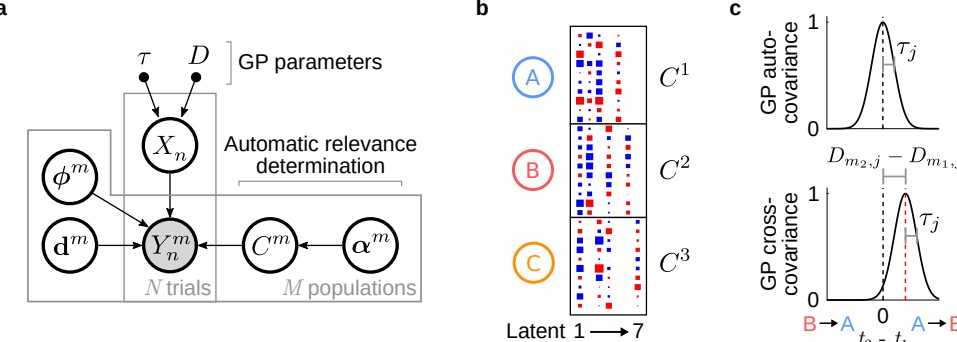

Figure 2: Delayed latents across multiple groups (mDLAG) (**a**) mDLAG directed graphical model representation. Filled circles represent observed variables. Unfilled circles represent probabilistic latent variables and parameters. Black dots represent point estimates. Arrows indicate conditional dependence relationships between variables. (**b**) Example mDLAG loading matrix (here the loading matrices for individual populations, $C^1$, $C^2$, and $C^3$ have been concatenated vertically). Each element of the matrix is represented by a square: magnitude is represented by the square's area, and sign is represented by the square's color (red: positive; blue: negative). Note the population-wise sparsity pattern of each latent variable, which is estimated from the neural activity. (**c**) Gaussian process auto-covariance (top) and cross-covariance (bottom) functions in the mDLAG model. Here we chose the squared exponential covariance function. The width of the auto- and cross-covariances corresponds to a timescale parameter ($\tau_j$). The center of the cross-covariance between populations $m_1$ and $m_2$ (e.g., population A and B, respectively) corresponds to the relative time delay $D_{m_2,j} - D_{m_1,j}$.

where the loading matrix $C^m \in \mathbb{R}^{q_m \times p}$ and mean parameter $\mathbf{d}^m \in \mathbb{R}^{q_m}$ are model parameters.[1] $\varepsilon^m$ is a zero-mean Gaussian random variable with noise precision matrix $\Phi^m \in \mathbb{S}^{q_m \times q_m}$ ($\mathbb{S}^{q_m \times q_m}$ is the set of $q_m \times q_m$ symmetric matrices). As we will describe, at time point $t$, latent variables $\mathbf{x}_{n,t}^m$, $m = 1, \ldots, M$ are coupled across populations, and thus each population has the same number of latents, $p$. Because we seek a low-dimensional description of neural activity, the number of latent variables is less than the number of neurons, i.e., $p < q$, where $q = \sum_m q_m$.

A core goal of the mDLAG framework is to identify and distinguish multiple network-level interactions (Fig. 1c). To do so requires identifying the number of latent variables across all recorded populations, and which subset of populations each latent involves, in a computationally tractable manner. We take a Bayesian approach to this problem (thereby avoiding computationally intensive grid search, see Discussion), and let $\mathbf{d}^m$, $\Phi^m$, and $C^m$ be probabilistic parameters with prior distributions.

The parameter $\mathbf{d}^m$ describes the mean firing rate of each neuron. We set a Gaussian prior over $\mathbf{d}^m$:

$$P(\mathbf{d}^m) = \mathcal{N}(\mathbf{d}^m \mid \mathbf{0}, \beta^{-1} I_{q_m}) \tag{3}$$

where $\beta \in \mathbb{R}_{>0}$ is a hyperparameter, and $I_{q_m}$ is the $q_m \times q_m$ identity matrix. We constrain the precision matrix $\Phi^m = \text{diag}(\phi_1^m, \ldots, \phi_{q_m}^m)$ to be diagonal to capture variance that is independent to each neuron. This constraint encourages the latent variables to explain as much of the shared variance among neurons as possible. We set the conjugate Gamma prior over each $\phi_i^m$, for each neuron $i = 1, \ldots, q_m$:

$$P(\phi_i^m) = \Gamma(\phi_i^m \mid a_\phi, b_\phi) \tag{4}$$

where $a_\phi, b_\phi \in \mathbb{R}_{>0}$ are hyperparameters.

The loading matrix $C^m$ linearly combines latent variables and maps them to observed neural activity. In particular, the $j^{\text{th}}$ column of $C^m$, $\mathbf{c}_j^m \in \mathbb{R}^{q_m}$, maps the $j^{\text{th}}$ latent variable $x_{n,j,t}^m$ to population $m$. To determine which subset of populations is described by each latent, we employ automatic relevance determination (ARD), which has been used successfully in a variety of contexts [17, 18]. Specifically, we define the following prior over the columns of each $C^m$:

$$P(\mathbf{c}_j^m \mid \alpha_j^m) = \mathcal{N}(\mathbf{c}_j^m \mid \mathbf{0}, (\alpha_j^m)^{-1} I_{q_m}) \tag{5}$$

$$P(\alpha_j^m) = \Gamma(\alpha_j^m \mid a_\alpha, b_\alpha) \tag{6}$$

---

[1]We will define all variables as they appear, but see Supplementary Section S1 for more complete notation.

where $\alpha_j^m \in \mathbb{R}_{>0}$ is the ARD parameter for latent variable $j$ and population $m$, and $a_\alpha, b_\alpha \in \mathbb{R}_{>0}$ are hyperparameters. As $\alpha_j^m$ becomes large, the magnitude of $\mathbf{c}_j^m$ becomes concentrated around 0, and hence the $j^{\text{th}}$ latent variable $x_{n,j,t}^m$ will have a vanishing influence on population $m$. The ARD prior encourages population-wise sparsity for each latent variable during model fitting (see below), where the loading matrix coefficients will be pushed toward zero for latent variables that explain an insignificant amount of shared variance within a population, and remain non-zero otherwise (Fig. 2b).

The parameter $C^m$ also has an intuitive geometric interpretation. Each element of $\mathbf{y}_{n,t}^m$, the activity of each neuron in population $m$, can be represented as an axis in a high-dimensional population activity space (Fig. 1b). Then each column of $C^m$ defines a dimension in population $m$'s activity space (Fig. 1b, "activity pattern"). Dimensions that appear in two or more populations represent patterns of activity that are correlated across populations. Note that the columns of $C^m$ are linearly independent; but they are not, in general, orthogonal. The ordering of these columns, and of the corresponding latent variables, is arbitrary.

**State model**  For each latent variable, we seek to characterize the direction of signal flow among the involved populations (determined by ARD) and how those signals evolve over time within and across trials. We therefore employ Gaussian processes (GPs) [19], and define a GP over all time points $t = 1, \ldots, T$ for each latent variable $j = 1, \ldots, p$ as follows (Fig. 2c):

$$\begin{bmatrix} \mathbf{x}_{n,j,:}^1 \\ \vdots \\ \mathbf{x}_{n,j,:}^M \end{bmatrix} \sim \mathcal{N}\left( \mathbf{0}, \begin{bmatrix} K_{1,1,j} & \cdots & K_{1,M,j} \\ \vdots & \ddots & \vdots \\ K_{M,1,j} & \cdots & K_{M,M,j} \end{bmatrix} \right) \tag{7}$$

Under equation 7, latents are independent and identically distributed across trials. The diagonal blocks $K_{1,1,j} = \cdots = K_{M,M,j} \in \mathbb{S}^{T \times T}$ describe the autocovariance of each latent, and each $T$-by-$T$ off-diagonal block describes the cross-covariance that couples two populations.

To define these matrices, we introduce additional notation. Specifically, we indicate populations with two subscripts, $m_1 = 1, \ldots, M$ and $m_2 = 1, \ldots, M$. Then, we define $K_{m_1, m_2, j} \in \mathbb{R}^{T \times T}$ to be either the auto- or cross-covariance matrix between latent variable $\mathbf{x}_{n,j,:}^{m_1} \in \mathbb{R}^T$ in population $m_1$ and latent variable $\mathbf{x}_{n,j,:}^{m_2} \in \mathbb{R}^T$ in population $m_2$ on trial $n$. We choose to use the squared exponential function for GP covariances (Fig. 2c). Therefore, element $(t_1, t_2)$ of each $K_{m_1, m_2, j}$ can be computed as follows [15, 20]:

$$k_{m_1, m_2, j}(t_1, t_2) = \left(1 - \sigma_j^2\right) \exp\left( -\frac{(\Delta t)^2}{2\tau_j^2} \right) + \sigma_j^2 \cdot \delta_{\Delta t} \tag{8}$$

$$\Delta t = (t_2 - D_{m_2, j}) - (t_1 - D_{m_1, j}) \tag{9}$$

where the characteristic timescale, $\tau_j \in \mathbb{R}_{>0}$, and the GP noise variance, $\sigma_j^2 \in (0, 1)$, are deterministic model parameters to be estimated from neural activity. $\delta_{\Delta t}$ is the kronecker delta, which is 1 for $\Delta t = 0$ and 0 otherwise.

We also introduce two new parameters: the time delay to population $m_1$, $D_{m_1, j} \in \mathbb{R}$, and the time delay to population $m_2$, $D_{m_2, j} \in \mathbb{R}$. Notice that, when computing the auto-covariance for population $m$ (i.e., when $m_1 = m_2 = m$; Fig. 2c, top), the time delay parameters $D_{m_1, j}$ and $D_{m_2, j}$ are equal, and so $\Delta t$ (equation 9) reduces simply to the time difference $(t_2 - t_1)$. Time delays are therefore only relevant when computing the cross-covariance between distinct populations $m_1$ and $m_2$. The time delay to population $m_1$, $D_{m_1, j}$, and the time delay to population $m_2$, $D_{m_2, j}$, by themselves have no physically meaningful interpretation. Their difference $D_{m_2, j} - D_{m_1, j}$, however, represents a well-defined, continuous-valued time delay from population $m_1$ to population $m_2$ (Fig. 2c, bottom). The sign of the relative time delay indicates the directionality of the lead-lag relationship between populations captured by latent variable $j$ (positive: population $m_1$ leads population $m_2$; negative: population $m_2$ leads population $m_1$), which we interpret as a description of signal flow. Note that time delays need not be integer multiples of the sampling period or spike count bin width of the neural activity.

Without loss of generality, we designate population $m = 1$ as the reference area, and fix the delays for population 1 at 0 (i.e., $D_{1,j} = 0$ for all latent variables $j = 1, \ldots, p$). We follow the same conventions as in [15, 21], and fix $\sigma_j^2$ to a small value ($10^{-3}$). Furthermore, the GP is normalized so

that $k_{m_1,m_2,j}(t_1, t_2) = 1$ if $\Delta t = 0$, thereby removing model redundancy in the scaling of the latent variables and loading matrices $C^m$.

**Posterior inference and fitting the mDLAG model**    Let $Y$ and $X$ be collections of all observed neural activity and latent variables, respectively, across all time points and trials. Similarly, let $\mathbf{d}$, $\phi$, $C$, $\mathcal{A}$, and $D$ be collections of the mean parameters, noise precisions, loading matrices, ARD parameters, and time delays, respectively. From the neural activity, we seek to estimate posterior distributions over the probabilistic model components $\theta = \{X, \mathbf{d}, \phi, C, \mathcal{A}\}$ and point estimates of the deterministic GP parameters $\Omega = \left\{ D, \{\tau_j\}_{j=1}^p \right\}$.

We do so by employing a variational inference scheme [16, 22], in which we maximize the evidence lower bound (ELBO), $L(Q, \Omega)$, with respect to the approximate posterior distribution $Q(\theta)$ and the deterministic parameters $\Omega$, where

$$\log P(Y) \geq L(Q, \Omega) = \mathbb{E}_Q[\log P(Y, \theta | \Omega)] - \mathbb{E}_Q[\log Q(\theta)] \tag{10}$$

We constrain $Q(\theta)$ so that it factorizes over the elements of $\theta$:

$$Q(\theta) = Q_x(X) Q_d(\mathbf{d}) Q_\phi(\phi) Q_c(C) Q_\mathcal{A}(\mathcal{A}) \tag{11}$$

This factorization enables closed-form updates during optimization. The ELBO can then be iteratively maximized via coordinate ascent of the factors of $Q(\theta)$ and the deterministic parameters $\Omega$. Here all hyperparameters were fixed to a very small value, $\beta, a_\phi, b_\phi, a_\alpha, b_\alpha = 10^{-12}$, to produce noninformative priors [16]. Throughout this work, we take estimates of the latent variables and model parameters to be the corresponding means of the posterior distributions comprising equation 11. Full details are provided in Supplementary Section S2.

**mDLAG special cases**    Finally, we consider some special cases of the mDLAG model that illustrate its relationship to other dimensionality reduction methods. In the case of two populations ($M = 2$), mDLAG is equivalent to a Bayesian formulation of DLAG [15]. In the case of one population ($M = 1$), and when all time delays are fixed to zero ($D_{m,j} = 0$), mDLAG becomes equivalent to a Bayesian formulation of Gaussian process factor analysis [23]. By removing temporal smoothing (i.e., in the limit as all GP noise parameters $\sigma_j$ approach 1) mDLAG becomes equivalent to GFA [16].

## 3    Validation in simulation and on neural recordings

**Simulation 1: Uncovering directed interactions across multiple populations**    We first sought to demonstrate the ability of mDLAG to infer the structure of multi-population interactions. We therefore generated simulated neural activity from three populations ($M = 3$) according to the mDLAG generative model (equations 1–9). For the sake of illustration, we set 10 neurons in each population ($q_m = 10$). Importantly, we designed the loading matrices $C^m$ so that all types of inter-population interactions were represented (Fig. 3a, left, ground truth): interactions shared globally, unique to each pair of populations, and local to one population. We also scaled the observation noise precision matrices $\Phi^m$ so that noise levels were representative of realistic neural activity. Specifically, activity due to single-neuron observation noise was 10 times stronger than activity due to latents (the "signal-to-noise ratio" $\text{tr}(C^m C^{m\top})/\text{tr}((\Phi^m)^{-1}) = 0.1$ for each population). We selected Gaussian process timescales and (relative) time delays that ranged between 20 ms to 150 ms and between 5 ms and 50 ms, respectively. With all model parameters specified, we then generated $N = 100$ independent and identically distributed trials. Each trial was 500 ms in length, comprising $T = 25$ time points with a sampling period of 20 ms, to mimic the 20 ms spike count time bins used to analyze the neural recordings in Section 4.

We then fit an mDLAG model to the simulated neural activity. We set the initial number of latents ($p = 10$) to be greater than the ground truth number ($p = 7$), to verify that these additional latents would be pruned during fitting. In addition, to demonstrate the benefit of ARD in the mDLAG model, we also fit a modified mDLAG model that did not use ARD. Specifically, we fit (via an exact EM algorithm) a modified model with state model defined by equations 7–9 and observation model defined by equations 1 and 2, but with no prior distributions over the parameters $C^m$, $\mathbf{d}^m$, and $\Phi^m$ (i.e., we obtained point estimates). The number of latents for this modified model was selected via 4-fold cross-validation, considering model candidates with $p = 1$ to $p = 10$ latents. Cross-validation is not needed for the mDLAG model with ARD, a key computational advantage of the approach.

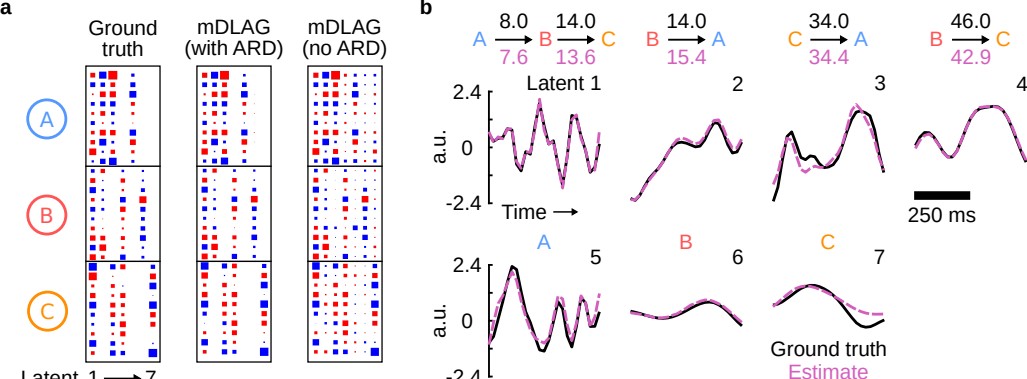

Figure 3: Automatic relevance determination encourages population-wise sparsity. (**a**) Loading matrix estimates. Left: Ground truth loading matrix. Center: mDLAG estimate. Right: mDLAG estimate where automatic relevance determination (ARD) was not used, i.e., no population-wise sparsity priors were placed on the loading matrix. Same conventions as in Fig. 2b. Note that the sign and ordering of each loading matrix column is, in general, arbitrary. We have therefore reordered and flipped the signs of the columns of the estimates to facilitate comparison with the ground truth. (**b**) Single-trial latent time course estimates. Each panel corresponds to the ground truth and estimated time course of a single latent variable. For concision, we show only latents corresponding to one population ($\mathbf{x}_{n,j,:}^{m}$); the remaining latents are time-shifted versions of those shown here. Inset above each latent are the involved populations along with the signal flow and magnitude of time delays between populations (in ms). Magenta: mDLAG estimates; black: ground truth. a.u.: arbitrary units.

We were able to identify the total number of latent variables ($p = 7$) with both approaches. The mDLAG model with ARD additionally recovered the population-wise sparsity structure with high accuracy (Fig. 3a, center). The mDLAG model without ARD, however, produced an estimate of the loading matrix with mostly non-zero elements (Fig. 3a, right): had we not known the ground truth in advance, it would be difficult to interpret which population subsets are involved in which interactions. The mDLAG model with ARD also estimated the latent variable time courses and time delays (along with their implied signal flow) with high accuracy (Fig. 3b; $R^2$ between ground truth and estimated time courses: 0.936; mean delay error: 1.14 ms).

**Simulation 2: Disentangling concurrent, bidirectional signaling** We next considered a simulated scenario that illustrates mDLAG's ability to distinguish concurrent, bidirectional signaling across populations, particularly in contrast with static multi-population methods like group factor analysis (GFA).[2] We again simulated activity from three populations, each with $q_m = 10$ neurons. These populations interacted through two latent processes, one that propagated from population A to B to C (Fig. 4a, Latent 1) and one that propagated in the opposite direction, from population C to B to A (Fig. 4a, Latent 2). Each latent incurred a 15 ms time delay propagating from one population to the next. For the sake of visual demonstration, we generated latent signals with a simple transient peak of activity (Fig. 4a, black traces), and generated the same latent signals on each trial. We then generated observed neural activity according to the mDLAG observation model (equations 1 and 2). Importantly, we scaled the columns of the loading matrices $C^m$ (for each area $m$) so that both columns had equal magnitude, and thus both latent interactions exhibited equal strength in each population: such a communication scheme is difficult to disentangle [15]. Finally, to isolate the consequences of model structure from issues like overfitting, we simulated a low-noise, data-rich scenario by generating $N = 1,000$ independent trials, each 500 ms in length (comprising $T = 25$ time points), with high signal-to-noise ratios $\operatorname{tr}(C^m C^{m\top})/\operatorname{tr}((\Phi^m)^{-1}) = 10.0$ for each population.

We fit both an mDLAG model and a GFA model (see Supplementary Section S3) to this simulated neural activity, assuming for each model the correct number of latent variables, $p = 2$. mDLAG latent variable and time delay estimates accurately reflected the distinct signaling pathways across

---

[2]See also Supplementary Fig. S2b for performance comparisons between GFA and mDLAG on the neural recordings analyzed in Section 4.

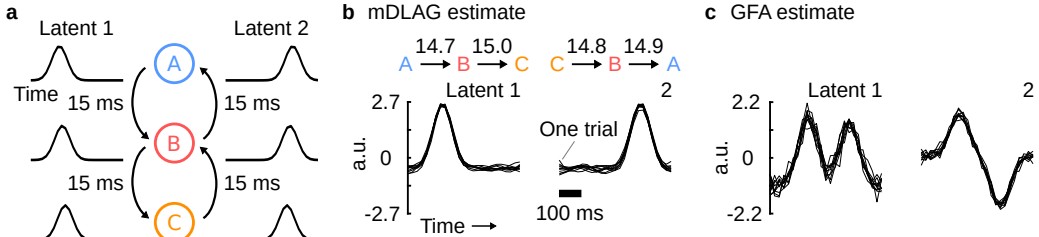

Figure 4: mDLAG disentangles concurrent signaling where static methods do not. (**a**) Schematic of simulated ground truth. Populations A, B, and C interact through two latents. Latent 1 (left) propagates from A to B to C. Latent 2 (right) propagates from C to B to A. (**b**) mDLAG estimates disentangle the two interactions. Each panel corresponds to a single latent variable. Each black trace represents one trial (10 representative trials shown). For concision, we show only latents corresponding to one population ($\mathbf{x}^m_{n,j,:}$); the remaining latents are time-shifted versions of those shown here. Inset above each latent are the involved populations along with the signal flow and magnitude of time delays between populations (in ms). a.u.: arbitrary units. (**c**) Group factor analysis (GFA) estimates represent mixtures of the two interactions. Same conventions as in (b).

the three populations (Fig. 4b; $R^2$ between ground truth and estimated time courses: 0.989; mean delay error: 0.16 ms). Each latent estimated by GFA, however, notably reflected a mixture of both interactions (Fig. 4c, each latent time course exhibits two peaks). With no description of signal flow, static methods like GFA struggle to distinguish concurrent, bidirectional signaling.

**Validating mDLAG on recordings from V1 and V2**    To further validate mDLAG, we turned to neural recordings from two areas in the macaque visual cortex, V1 and V2 [24]. These recordings have been studied extensively [15, 25–27], and therefore provide an excellent testbed for our framework. We applied mDLAG to the simultaneously recorded activity of neuronal populations in the superficial (output) layers of V1 (61 to 122 neurons; mean 86.3), and the middle (input) layers of V2 (15 to 32 neurons; mean 19.6) in three anesthetized animals (Supplementary Fig. S1a). We analyzed neuronal responses measured during the 1.28 second presentation of drifting sinusoidal gratings of different orientations, and counted spikes in 20 ms time bins. Because we were interested in V1-V2 interactions on timescales within a trial, we subtracted the mean across time bins within each trial from each neuron. This step removed activity that fluctuated on slow timescales from one stimulus presentation to the next [28]. In total, we analyzed separately 40 "datasets," corresponding to five recording sessions, each with eight different orientations. Each dataset included 400 trials: we allocated at random 300 trials as a training set and 100 trials as a test set on which to measure model performance (we employed a leave-group-out prediction metric; see Supplementary Section S4).

Given prior work on these V1-V2 recordings, we reasoned that mDLAG ought to qualitatively recover several hallmarks of the V1-V2 activity: (1) a significant number of latent variables local to each area, indicative of selective V1-V2 communication (Supplementary Fig. S1b, 'V1' and 'V2'; Supplementary Fig. S1c, latents 4 and 5), (2) the presence of periodic structure representative of the drifting grating stimulus (Supplementary Fig. S1c, latents 3 and 4), and (3) bidirectional interactions (Supplementary Fig. S1c, latents 1–3). Indeed, mDLAG detected all of these hallmarks.

These recordings were also previously studied with DLAG [15], which was designed to study concurrent signaling between two neuronal populations. Since DLAG is a special case of the mDLAG framework, we reasoned that mDLAG ought to exhibit similar quantitative performance and extract qualitatively similar features of V1-V2 communication. Indeed, the latent time courses estimated by mDLAG (Supplementary Fig. S1c) were qualitatively similar to those estimated by DLAG [29] (Supplementary Fig. S1d). Interestingly, mDLAG outperformed DLAG across all datasets (Supplementary Fig. S2a, points above the diagonal), suggesting that ARD provides an improved method of model selection over the constrained grid search method used for DLAG, while also avoiding grid search's computational drawbacks (see Supplementary Section S5). Collectively, the results across our simulations and the V1-V2 recordings indicate that mDLAG is well-equipped to study concurrent signaling across multiple simultaneously recorded neuronal populations.

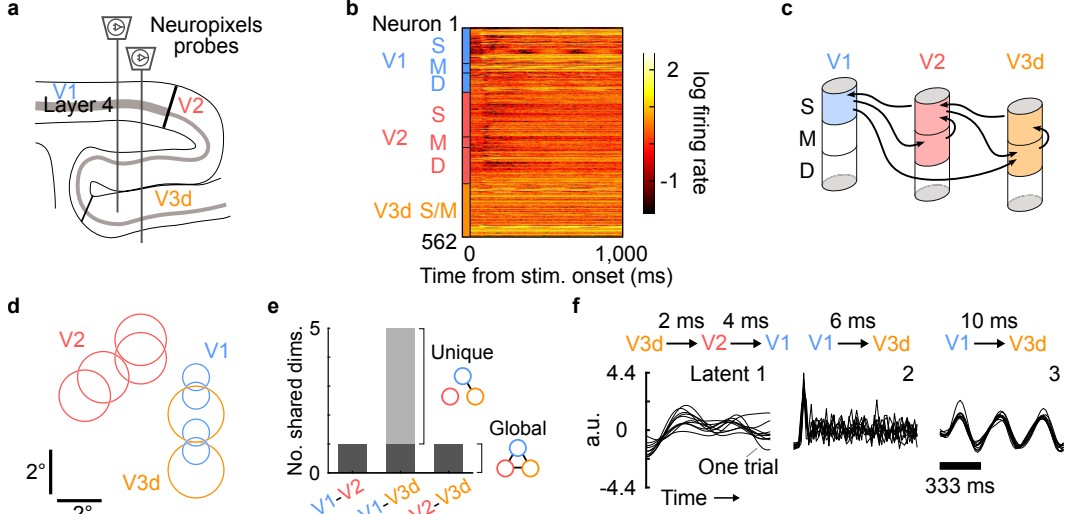

Figure 5: Applying mDLAG to Neuropixels recordings across V1, V2, and V3d. (**a**) Schematic showing a sagittal section of occipital cortex and the recording setup. Laminar compartments of V1, V2, and V3d were recorded using multiple Neuropixels probes. (**b**) Temporally smoothed peristimulus time histograms during the stimulus presentation period, for an example session and stimulus condition. (**c**) Schematic showing the subset of laminar compartments included in the analyses (shaded compartments). Compartments with the same color were treated as one population during analysis. Arrows indicate prominent anatomical connections [30]. (**d**) Visual receptive field locations of V1, V2, and V3d. Each circle represents receptive field size and location across the channels of a single Neuropixels probe. Scale bars indicate degrees of visual field. (**e**) Total number of latent dimensions shared between each pair of populations. Light gray: dimensions unique to a pair. Dark gray: dimensions shared globally across all populations. (**f**) Single-trial time course estimates of select shared latents (same dataset as in (b), (d), and (e)). Same conventions as in Fig. 4b.

## 4 Interactions across laminar compartments of V1, V2, and V3d

We next used mDLAG to study interactions across V1, V2, and V3d of anesthetized macaques recorded using multiple Neuropixels probes (Fig. 5a). These recordings included hundreds of neurons across different layers in each area (Fig. 5b, superficial (S), middle (M), and deep (D) compartments) allowing us to investigate longstanding questions about functional cortical organization.

First, we sought to quantify the performance of mDLAG models relative to alternative approaches, and to further demonstrate the empirical benefit of mDLAG's model components. We therefore considered three types of models: group factor analysis (GFA); mDLAG, but with time delays removed ('mDLAG-0'); and mDLAG. With each model type, we analyzed spike counts (in 20 ms bins) of neurons in select laminar compartments of each area, measured during the first 1,000 ms after the onset of a drifting sinusoidal grating (Fig. 5b, trial-averaged responses for an example dataset). The selected laminar compartments (grouped into three sub-populations for analysis: Fig. 5c, color shading) were those that likely directly interacted through a combination of feedforward, feedback, and inter-laminar connections [30] (Fig. 5c, black arrows). As we did for the V1-V2 recordings, above, we subtracted the mean across time bins within each trial from each neuron, to remove slow fluctuations beyond the timescale of a trial. We fit GFA, mDLAG-0, and mDLAG separately to 10 datasets, comprising five recording sessions (each with two grating orientations, 90° apart, presented 300 trials each) from two anesthetized animals. For each dataset, we allocated at random 225 trials as a training set and 75 trials as a test set on which to measure model performance. GFA, mDLAG-0, and mDLAG exhibited increasingly better leave-group-out prediction (Supplementary Fig. S2b: mDLAG-0 better than GFA; c: mDLAG better than mDLAG-0), demonstrating the performance benefit of (1) temporal smoothing and (2) time delays (see also Supplementary Fig. S3 for a demonstration of mDLAG performance versus number of available training trials, and Supplementary Fig. S4 for a characterization of GFA and mDLAG runtimes).

We then focused on one particular set of questions: Do V1, V2, and V3d communicate selectively, and if so, what aspects of the recorded populations might contribute to this selectivity? One likely aspect is the alignment of visual receptive fields (RFs): retinotopically aligned neuronal populations might co-contribute to computations that non-aligned populations would not. This functional organization would appear as mDLAG latent variables shared exclusively between the aligned populations. We therefore studied a dataset from a recording session in which the retinotopy of the recorded populations allowed us to investigate this hypothesis (Fig. 5d). RFs of the V1 and V3d populations were largely overlapping (Fig. 5d, blue: V1, gold: V3d), whereas the RFs in V2 did not overlap with either V1 or V3d (Fig. 5d, red: V2).

We quantified the total number of dimensions shared between each pair of populations (Fig. 5e; a dimension was considered significant in a population if it explained at least 2% of the shared variance, Supplementary Section S5, equation S52). mDLAG allowed us to distinguish dimensions that were unique to a pair of populations (e.g., between V1 and V3d, but not V2) from dimensions that also involved the third population. V1-V3d interactions included unique dimensions (Fig. 5e, 'V1-V3d', light gray) in addition to a dimension shared globally across all three populations (Fig. 5e, 'V1-V3d', dark gray), whereas V1-V2 interactions and V2-V3d interactions could be attributed primarily to the global dimension (Fig. 5e, 'V1-V2' and 'V2-V3d', only dark gray). The single-trial time courses of the global latent evolved according to a long timescale (Fig. 5f, latent 1, 132 ms) relative to the unique V1-V3d latents (Fig. 5f, latents 2 and 3, 8 ms and 62 ms, respectively). The global latent was also associated with a direction of signal flow (V3d to V2 to V1) opposite to that of the V1-V3d latents (V1 to V3d). The time courses of the V1-V3d latents exhibited features of the stimulus response, including a fast transient response (Fig. 5f, latent 2) and periodic structure with the same period as the drifting grating stimulus (Fig. 5f, latent 3, 333 ms period). Results were largely insensitive to the initialization of the mDLAG fitting procedure (Supplementary Fig. S5). These findings are consistent with the hypothesis that visual cortical populations communicate selectively in a retinotopic manner, as they perform computations on inputs from localized regions of visual space [31].

## 5 Discussion

Even for two populations, identifying which latents are shared between populations or local to a population is a challenging computational problem. Existing approaches explicitly designate latents as shared or local [13–15], and then rely on heuristics to avoid the computational cost of grid search. Scaling this type of approach to three or more populations (or external experimental variables) would be prohibitive. mDLAG is thus an advance toward scaling to large-scale multi-population recordings, and could continue to be improved through the many approaches used to scale Gaussian process methods in neuroscience [13, 23, 32–36].

mDLAG treats time delays as constant parameters across trials, time, and neurons. The delay estimated for each latent variable thus represents a summary of inter-population interaction throughout the course of an experiment. To assess the variability of delay estimates, one could fit mDLAG to subsets of trials, subsets of neurons, or to separate trial epochs. If additional subpopulation labels are available (for example, laminar compartments in our Neuropixels recordings, Fig. 5b), one could also specify these subpopulations to the mDLAG model to better incorporate variability of delays across neurons.

A latent variable shared across three or more populations is potentially consistent with multiple signaling schemes. For instance, Latent 1 of Fig. 5f is consistent with a signal relayed from V3d to V2 (with a delay of 2 ms) and then from V2 to V1 (with a delay of 4 ms). It is also consistent, however, with a scheme in which V3d is a common input to V2 (with a delay of 2 ms) and to V1 (with a delay of 6 ms). A third scheme could involve common input to all three areas from an unobserved source. Still, mDLAG narrows the set of populations that could be involved in any given interaction, and the sign and magnitude of mDLAG's time delays narrow the set of signaling schemes consistent with the data. This hypothesis set can be narrowed further by experimental interventions [37].

The mDLAG model includes assumptions of linearity and temporal smoothness; specifically, we have employed here the commonly used squared exponential function for mDLAG's GP covariances. It might be desirable to incorporate temporal structure more appropriate for certain signals (for example, the sinusoidal structure of Latent 3, Fig. 5f). Then, an alternative GP covariance function can be substituted in a straightforward manner into the mDLAG model (equation 8) [19, 32, 33, 38].

Nonlinearities, for example related to neuronal tuning or receptive fields, could also be incorporated under a GP framework [39, 40].

For exploratory data analysis, mDLAG's GP-based description of multi-population temporal structure is advantageous over an alternative linear dynamical system (LDS)-based description [10, 11] in two respects: (1) a GP can be useful for exploratory data analyses where an appropriate parametric dynamical model is unknown *a priori*, and (2) mDLAG's continuous-time model enables the discovery of wide-ranging delays with high precision, which, in contrast to discrete-time LDS approaches, are not limited to be integer multiples of the sampling period or spike count bin width of the neural activity. Ultimately, these approaches can be complementary: one can use mDLAG to generate data-driven hypotheses about motifs of concurrent signaling across populations, and then test these hypotheses with a dynamical system-based approach.

Finally, while we applied mDLAG to the spiking activity of neuronal populations, mDLAG is applicable to any high-dimensional time series data, including other neural recording modalities. In fact, the groups analyzed by mDLAG need not all be neuronal populations, but could include, for example, a collection of dynamic stimulus or behavioral variables. mDLAG is a general framework that advances the study of concurrent signaling throughout the brain.

## Code availability

A MATLAB (MathWorks) implementation of mDLAG is available on GitHub at `http://github.com/egokcen/mDLAG` and on Zenodo at `https://doi.org/10.5281/zenodo.10048163` [41].

## Acknowledgments

We thank A. Zandvakili for providing the V1-V2 recordings in Supplementary Fig. S1. This work was supported by the Dowd Fellowship (E.G.), ECE Benjamin Garver Lamme/Westinghouse Fellowship (E.G.), Simons Collaboration on the Global Brain 542999 (A.K.), 543009 (C.K.M.), 543065 (B.M.Y.), NIH RF1 NS127107 (A.K., C.K.M., B.M.Y.), NIH R01 EY035896 (B.M.Y., A.K., C.K.M.), NIH U01 NS094288 (C.K.M.), NIH CRCNS R01 MH118929 (B.M.Y.), NSF NCS DRL 2124066 (B.M.Y.), NIH R01 NS129584 (B.M.Y.).

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
