# Uncovering motifs of concurrent signaling across multiple neuronal populations
# — Supplementary Information —

**Evren Gokcen**[1], **Anna I. Jasper**[2], **Alison Xu**[2],
**Adam Kohn**[*,2,3,4], **Christian K. Machens**[*,5], **Byron M. Yu**[*,1,6]
[1]Dept. of Electrical and Computer Engineering, Carnegie Mellon University
[2]Dominick Purpura Dept. of Neuroscience, [3]Dept. of Ophthalmology and Visual Sciences,
[4]Dept. of Systems and Computational Biology, Albert Einstein College of Medicine
[5]Champalimaud Neuroscience Programme, Champalimaud Foundation
[6]Dept. of Biomedical Engineering, Carnegie Mellon University
egokcen@cmu.edu, {anna.jasper, alison.xu, adam.kohn}@einsteinmed.edu,
christian.machens@neuro.fchampalimaud.org, byronyu@cmu.edu
[*]Denotes equal contribution.

## S1   Mathematical notation

To disambiguate each variable or parameter in the mDLAG model, we need to keep track of up to four labels that indicate their associated (1) trial; (2) neuron or latent variable index; (3) time point; or (4) subpopulation (for example, brain area). We indicate the first three labels via subscripts. Trials are indexed by $n = 1, \ldots, N$; neurons are indexed by $i = 1, \ldots, q$; latent variables are indexed by $j = 1, \ldots, p$; and time is indexed by $t = 1, \ldots, T$. Where relevant, we indicate the population to which a variable or parameter pertains via a superscript, where populations are indexed by $m = 1, \ldots, M$. For example, we define the observed activity of neuron $i$ (out of $q_m$) in population $m$ at time $t$ on trial $n$ as $y^m_{n,i,t} \in \mathbb{R}$. Similarly, we define latent variable $j$ (out of $p$) in population $m$ at time $t$ on trial $n$ as $x^m_{n,j,t} \in \mathbb{R}$. To indicate a collection of all variables along a particular index, we replace that index with a colon. Hence we represent the simultaneous activity of $q_m$ neurons observed in population $m$ at time $t$ on trial $n$ as the vector $\mathbf{y}^m_{n,:,t} \in \mathbb{R}^{q_m}$. Similarly, we represent the collection of all $p$ latent variables in population $m$ at time $t$ on trial $n$ as the vector $\mathbf{x}^m_{n,:,t} \in \mathbb{R}^p$. For concision, where a particular index is either not applicable or not immediately relevant, we omit it. The identities of the remaining indices should be clear from context. For example, we might rewrite $\mathbf{y}^m_{n,:,t}$ as $\mathbf{y}^m_{n,t}$.

It is conceptually helpful to understand the notation for observed variables ($\mathbf{y}$) and latent variables ($\mathbf{x}$) as taking cross-sections of three-dimensional arrays. For example, observed activity in population $m$ on trial $n$ can be grouped into the matrix (two-dimensional array) $Y^m_n = [\mathbf{y}^m_{n,1} \cdots \mathbf{y}^m_{n,T}] \in \mathbb{R}^{q_m \times T}$. Hence each $\mathbf{y}^m_{n,t}$ is a column of $Y^m_n$. Then we can form the three-dimensional array $Y^m$ by concatenating the matrices $Y^m_1, \ldots, Y^m_N$ across trials along a third dimension. Similarly, the latent variables in population $m$ on trial $n$ can be grouped into the matrix $X^m_n = [\mathbf{x}^m_{n,:,1} \cdots \mathbf{x}^m_{n,:,T}] \in \mathbb{R}^{p \times T}$. We represent a row of $X^m_n$ (i.e., the values of a single latent variable $j$ at all time points on trial $n$) as $\mathbf{x}^m_{n,j,:} \in \mathbb{R}^T$. Finally, we can form the three-dimensional array $X^m$ by concatenating the matrices $X^m_1, \ldots, X^m_N$ across trials along a third dimension.

We will explicitly define all other variables and parameters as they appear, but for reference, we list common variables and parameters below:

**Data characteristics**

- $N$ — number of trials
- $T$ — number of time points per trial

**Observed neural activity**

- $q_m$ — number of neurons observed in population $m$
- $Y_n^m$ — $q_m \times T$ matrix of observed activity in population $m$ on trial $n$
- $\mathbf{y}_{n,t}^m$ — $q_m \times 1$ vector of observed activity in population $m$ at time $t$ on trial $n$; the $t^{\text{th}}$ column of $Y_n^m$

**Latent variables**

- $p$ — number of latent variables (same for all populations)
- $X_n^m$ — $p \times T$ matrix of latent variables in population $m$ on trial $n$
- $\mathbf{x}_{n,:,t}^m$ — $p \times 1$ vector of latent variables in population $m$ at time $t$ on trial $n$; the $t^{\text{th}}$ column of $X_n^m$
- $\mathbf{x}_{n,j,:}^m$ — $T \times 1$ vector of values of latent $j$ in population $m$ over time on trial $n$; the $j^{\text{th}}$ row of $X_n^m$

**Observation model parameters**

- $C^m$ — $q_m \times p$ loading matrix for population $m$
- $\alpha_j^m$ — automatic relevance determination (ARD) parameter for population $m$ and latent $j$
- $\mathbf{d}^m$ — $q_m \times 1$ mean parameter for population $m$
- $\phi^m$ — $q_m \times 1$ observation noise precision parameter for population $m$, $\phi^m = [\phi_1^m \cdots \phi_{q_m}^m]^\top$

**Gaussian process parameters**

- $D_{m,j}$ — time delay parameter between population $m$ and latent $j$
- $\tau_j$ — Gaussian process timescale for latent $j$
- $\sigma_j$ — Gaussian process noise parameter for latent $j$

**Gaussian process covariances**

- $K_{m_1,m_2,j}$ — $T \times T$ covariance matrix for latent $j$, between populations $m_1$ and $m_2$
- $k_{m_1,m_2,j}$ — covariance function for latent $j$, between populations $m_1$ and $m_2$

**Hyperparameters (fixed to small values to produce noninformative priors)**

- $\beta$ — precision parameter of the Gaussian prior over each mean parameter $\mathbf{d}^m$
- $a_\phi,\ b_\phi$ — shape and rate parameters, respectively, of the Gamma prior over each noise precision parameter $\phi_i^m$ for neuron $i$ in population $m$
- $a_\alpha,\ b_\alpha$ — shape and rate parameters, respectively, of the Gamma prior over each ARD parameter $\alpha_j^m$ for population $m$ and latent $j$

## S2   Posterior inference and fitting the mDLAG model

### S2.1   Variational inference

Let $Y$ and $X$ be collections of all observed neural activity and latent variables, respectively, across all time points and trials. Similarly, let $\mathbf{d}$, $\phi$, $C$, $\mathcal{A}$, and $D$ be collections of the mean parameters,

noise precisions, loading matrices, ARD parameters, and time delays, respectively. From the neural activity, we seek to estimate posterior distributions over the probabilistic model components

$$\theta = \{X, \, \mathbf{d}, \, \boldsymbol{\phi}, \, C, \, \mathcal{A}\} \tag{S1}$$

and point estimates of the deterministic GP parameters $\Omega = \left\{ D, \, \{\tau_j\}_{j=1}^p \right\}$.

In the case of methods like GPFA [1] and DLAG [2], the linear-Gaussian structure of the model enables an exact expectation-maximization (EM) algorithm. With the introduction of prior distributions over model parameters, mDLAG loses this property. The complete likelihood of the mDLAG model,

$$
\begin{aligned}
P(Y, \theta | \Omega) &= P(\mathbf{d}) P(\boldsymbol{\phi}) P(C | \mathcal{A}) P(\mathcal{A}) P(Y | X, C, \mathbf{d}, \boldsymbol{\phi}) P(X | \Omega) \\
&= \prod_{m=1}^{M} \left[ P(\mathbf{d}^m) \left[ \prod_{i=1}^{q_m} P(\phi_i^m) \right] \left[ \prod_{j=1}^{p} P(\mathbf{c}_j^m \mid \alpha_j^m) P(\alpha_j^m) \right] \right. \\
&\quad \left. \cdot \left[ \prod_{n=1}^{N} \prod_{t=1}^{T} P(\mathbf{y}_{n,t}^m | \mathbf{x}_{n,t}^m, C^m, \mathbf{d}^m, \boldsymbol{\phi}^m) \right] \right] \cdot \left[ \prod_{n=1}^{N} \prod_{j=1}^{p} P(\mathbf{x}_{n,j,:} | \{D_{m,j}\}_{m=1}^M, \tau_j) \right] \quad \text{(S2)}
\end{aligned}
$$

is no longer Gaussian. Then a hypothetical EM E-step (evaluation of the posterior distribution $P(\theta | Y, \Omega)$) becomes prohibitive, as it relies on the analytically intractable marginalization of equation S2 with respect to $\theta$.

We therefore employ instead a variational inference scheme [3, 4], in which we maximize the evidence lower bound (ELBO), $L(Q, \Omega)$, with respect to the approximate posterior distribution $Q(\theta)$ and the deterministic parameters $\Omega$, where

$$\log P(Y) \geq L(Q, \Omega) = \mathbb{E}_Q[\log P(Y, \theta | \Omega)] - \mathbb{E}_Q[\log Q(\theta)] \tag{S3}$$

We constrain $Q(\theta)$ so that it factorizes over the elements of $\theta$:

$$Q(\theta) = Q_x(X) Q_d(\mathbf{d}) Q_\phi(\boldsymbol{\phi}) Q_c(C) Q_\mathcal{A}(\mathcal{A}) \tag{S4}$$

This factorization enables closed-form updates during optimization (see below). The ELBO can then be iteratively maximized via coordinate ascent of the factors of $Q(\theta)$ and the deterministic parameters $\Omega$: each factor or deterministic parameter is updated in turn while the remaining factors or parameters are held fixed. These updates are repeated until the ELBO, which is guaranteed to be non-decreasing, improves from one iteration to the next by less than a present tolerance (here we used $10^{-8}$ or a maximum of 50,000 iterations).

### S2.1.1 Posterior distribution updates

Maximizing the ELBO, $L(Q, \Omega)$, with respect to the $k^{\text{th}}$ factor of $Q$, $Q_k^*$, results in the following update [3]:

$$\log Q_k^*(\theta_k) = \langle \log P(Y, \theta | \Omega) \rangle_{\ell \neq k} + \text{const.} \tag{S5}$$

Here we introduce the notation $\langle \cdot \rangle$ to indicate the expectation with respect to the approximate posterior distribution, $\mathbb{E}_Q[\cdot]$, and $\langle \log P(Y, \theta | \Omega) \rangle_{\ell \neq k}$ specifically indicates the expectation of the complete log likelihood with respect to all but the $k^{\text{th}}$ factor of $Q$. We impose no further constraints on $Q$ or its factors. However, because of the choice of Gaussian and conjugate Gamma priors in Section 2 of the main text, evaluation of equation S5 leads to factors with the same functional form as their

corresponding priors (equations 1–7):

$$Q_x(X) = \prod_{n=1}^{N} \mathcal{N}(\bar{\mathbf{x}}_n \mid \bar{\boldsymbol{\mu}}_{x_n}, \bar{\Sigma}_x) \tag{S6}$$

$$Q_d(\mathbf{d}) = \prod_{m=1}^{M} \mathcal{N}(\mathbf{d}^m \mid \boldsymbol{\mu}_d^m, \Sigma_d^m) \tag{S7}$$

$$Q_\phi(\boldsymbol{\phi}) = \prod_{m=1}^{M} \prod_{i=1}^{q_m} \Gamma(\phi_i^m \mid \widetilde{a}_\phi, \widetilde{b}_{\phi,i}^m) \tag{S8}$$

$$Q_c(C) = \prod_{m=1}^{M} \prod_{i=1}^{q_m} \mathcal{N}(\widetilde{\mathbf{c}}_i^m \mid \widetilde{\boldsymbol{\mu}}_{c_i}^m, \Sigma_{c_i}^m) \tag{S9}$$

$$Q_{\mathcal{A}}(\mathcal{A}) = \prod_{m=1}^{M} \prod_{j=1}^{p} \Gamma(\alpha_j^m \mid \widetilde{a}_\alpha^m, \widetilde{b}_{\alpha,j}^m) \tag{S10}$$

Here $\bar{\mathbf{x}}_n \in \mathbb{R}^{MpT}$ is a collection of all latent variables on trial $n$ (see below), and $\widetilde{\mathbf{c}}_i^m \in \mathbb{R}^p$ is the $i^{\text{th}}$ row of $C^m$, the loading matrix for population $m$. Any additional factorization in equations S6–S10 also emerge naturally—we impose only the factorization in equation S4.

To express the updates for $Q_x(X)$, let us first define several variables. Construct $\mathbf{y}_{n,t} = [\mathbf{y}_{n,t}^{1\top} \cdots \mathbf{y}_{n,t}^{M\top}]^\top \in \mathbb{R}^q$, $q = \sum_m q_m$, by vertically concatenating the neural activity of populations $m = 1, \ldots, M$ at time $t$ on trial $n$. Then construct $\bar{\mathbf{y}}_n = [\mathbf{y}_{n,1}^\top \cdots \mathbf{y}_{n,T}^\top]^\top \in \mathbb{R}^{qT}$ by vertically concatenating the neural activity $\mathbf{y}_{n,t}$ across all time points $t = 1, \ldots, T$. For latent variables, define $\mathbf{x}_{n,t} = [\mathbf{x}_{n,:,t}^{1\top} \cdots \mathbf{x}_{n,:,t}^{M\top}]^\top \in \mathbb{R}^{Mp}$ by vertically concatenating the $p$ latent variables of each population at time $t$ on trial $n$. Then we vertically concatenate the latent variables $\mathbf{x}_{n,t}$ across all time points $t = 1, \ldots, T$ to give $\bar{\mathbf{x}}_n = [\mathbf{x}_{n,1}^\top \cdots \mathbf{x}_{n,T}^\top]^\top \in \mathbb{R}^{MpT}$. Finally, we collect the parameters $C^m$, $\Phi^m$, and $\mathbf{d}^m$ across populations $m = 1, \ldots, M$ by defining $C = \text{diag}(C^1, \ldots, C^M) \in \mathbb{R}^{q \times Mp}$, $\Phi = \text{diag}(\Phi^1, \ldots, \Phi^m) \in \mathbb{S}^{q \times q}$, and $\mathbf{d} = [\mathbf{d}^{1\top} \cdots \mathbf{d}^{M\top}]^\top \in \mathbb{R}^q$.

Posterior estimates of the latent variables $X$ are independent across trials. We can thus update $Q_x(X)$ by evaluating the posterior covariance, $\bar{\Sigma}_x \in \mathbb{S}^{MpT \times MpT}$, and mean, $\bar{\boldsymbol{\mu}}_{x_n} \in \mathbb{R}^{MpT}$, of $\bar{\mathbf{x}}_n$ for each trial $n$:

$$\bar{\Sigma}_x = \left(\bar{K}^{-1} + \langle \overline{C^\top \Phi C} \rangle \right)^{-1} \tag{S11}$$

$$\bar{\boldsymbol{\mu}}_{x_n} = \bar{\Sigma}_x \langle \bar{C} \rangle^\top \langle \bar{\Phi} \rangle \left(\bar{\mathbf{y}}_n - \langle \bar{\mathbf{d}} \rangle \right) \tag{S12}$$

where $\langle \bar{C} \rangle \in \mathbb{R}^{qT \times MpT}$, $\langle \bar{\Phi} \rangle \in \mathbb{S}^{qT \times qT}$, and $\langle \overline{C^\top \Phi C} \rangle \in \mathbb{R}^{MpT \times MpT}$ are block diagonal matrices comprising $T$ copies of the matrices $\langle C \rangle$, $\langle \Phi \rangle$, and $\langle C^\top \Phi C \rangle$, respectively. $\langle \bar{\mathbf{d}} \rangle \in \mathbb{R}^{qT}$ is constructed by vertically concatenating $T$ copies of $\langle \mathbf{d} \rangle$. The elements of $\bar{K} \in \mathbb{R}^{MpT \times MpT}$ are computed using equations 8 and 9. Note that (1) The update for the posterior covariance, $\bar{\Sigma}_x$, is identical for trials of the same length. This computation can therefore be reused efficiently across trials. (2) Under the posterior distribution, latent variables $j = 1, \ldots, p$ are no longer independent, as they are under the prior distribution (equations 7, S2).

Posterior estimates of the mean parameters $\mathbf{d}$ are independent across populations (and, in fact, neurons). We can thus update $Q_d(\mathbf{d})$ by evaluating the posterior covariance, $\Sigma_d^m \in \mathbb{S}^{q_m \times q_m}$, and mean, $\boldsymbol{\mu}_d^m \in \mathbb{R}^{q_m}$, of mean parameter $\mathbf{d}^m$ for each population $m$:

$$\Sigma_d^m = \left(\beta I_{q_m} + NT \langle \Phi^m \rangle \right)^{-1} \tag{S13}$$

$$\boldsymbol{\mu}_d^m = \Sigma_d^m \langle \Phi^m \rangle \sum_{n=1}^{N} \sum_{t=1}^{T} \left(\mathbf{y}_{n,t}^m - \langle C^m \rangle \langle \mathbf{x}_{n,t}^m \rangle \right) \tag{S14}$$

Posterior estimates of precision parameters $\phi$ are independent across populations and neurons. We can thus update $Q_\phi(\boldsymbol{\phi})$ by evaluating the posterior parameters $\widetilde{a}_\phi$ and $\widetilde{b}_{\phi,i}^m$ of parameter $\phi_i^m$ for each

neuron $i$ in population $m$:

$$\widetilde{a}_\phi = a_\phi + \frac{NT}{2} \tag{S15}$$

$$\widetilde{b}_{\phi,i}^m = b_\phi + \frac{1}{2} \sum_{n=1}^N \sum_{t=1}^T \left[ (y_{n,i,t}^m)^2 + \langle (d_i^m)^2 \rangle + \text{tr} \left( \langle \widetilde{\mathbf{c}}_i^m (\widetilde{\mathbf{c}}_i^m)^\top \rangle \langle \mathbf{x}_{n,t}^m (\mathbf{x}_{n,t}^m)^\top \rangle \right) \right.$$

$$\left. - 2 \langle \widetilde{\mathbf{c}}_i^m \rangle \langle \mathbf{x}_{n,t}^m \rangle \left( y_{n,i,t}^m - \langle d_i^m \rangle \right) - 2 y_{n,i,t}^m \langle d_i^m \rangle \right] \tag{S16}$$

Here $\widetilde{\mathbf{c}}_i^m \in \mathbb{R}^p$ is again the $i^{\text{th}}$ row of $C^m$, the loading matrix for population $m$.

Posterior estimates of loading matrices $C$ are independent across populations and neurons, i.e., across the rows of each $C^m$. We can thus update $Q_c(C)$ by evaluating the posterior covariance, $\Sigma_{c_i}^m \in \mathbb{S}^{p \times p}$, and mean, $\widetilde{\boldsymbol{\mu}}_{c_i}^m \in \mathbb{R}^p$, of the $i^{\text{th}}$ row of $C^m$:

$$\Sigma_{c_i}^m = \left( \langle \mathcal{A}^m \rangle + \langle \phi_i^m \rangle \sum_{n=1}^N \sum_{t=1}^T \langle \mathbf{x}_{n,t}^m (\mathbf{x}_{n,t}^m)^\top \rangle \right)^{-1} \tag{S17}$$

$$\widetilde{\boldsymbol{\mu}}_{c_i}^m = \Sigma_{c_i}^m \langle \phi_i^m \rangle \sum_{n=1}^N \sum_{t=1}^T \langle \mathbf{x}_{n,t}^m \rangle \left( y_{n,i,t}^m - \langle d_i^m \rangle \right) \tag{S18}$$

Here $\mathcal{A}^m = \text{diag}(\alpha_1^m, \ldots, \alpha_p^m)$. Note that the posterior independence over the rows of each $C^m$ contrasts with the prior independence over the columns of each $C^m$ (equations 5, S2).

Finally, posterior estimates of ARD parameters $\mathcal{A}$ are independent across populations and latent variables. We can thus update $Q_{\mathcal{A}}(\mathcal{A})$ by evaluating the posterior parameters $\widetilde{a}_\alpha^m$ and $\widetilde{b}_{\alpha,j}^m$ of parameter $\alpha_j^m$ for each population $m$ and latent variable $j$:

$$\widetilde{a}_\alpha^m = a_\alpha + \frac{q_m}{2} \tag{S19}$$

$$\widetilde{b}_{\alpha,j}^m = b_\alpha + \frac{1}{2} \langle \|\mathbf{c}_j^m\|_2^2 \rangle \tag{S20}$$

All moments $\langle \cdot \rangle$ can be readily computed from the approximate posterior distributions given in equations S6–S10.

### S2.1.2 Gaussian process parameter updates

There are no closed-form solutions for the Gaussian process parameter updates, but we can compute gradients and perform gradient ascent. Note that, for this work, we choose not to fit the Gaussian process noise variances $\sigma_j^2$, but rather, we set them to small values ($10^{-3}$), as in [1, 2].

To express the timescale and delay parameter gradients, we introduce more compact notation for the variables in equation 7. Let $\mathbf{x}_{n,j,:} = [\mathbf{x}_{n,j,:}^{1\top} \cdots \mathbf{x}_{n,j,:}^{M\top}]^\top \in \mathbb{R}^{MT}$ for the $j^{\text{th}}$ latent, and

$$K_j = \begin{bmatrix} K_{1,1,j} & \cdots & K_{1,M,j} \\ \vdots & \ddots & \vdots \\ K_{M,1,j} & \cdots & K_{M,M,j} \end{bmatrix} \in \mathbb{S}^{MT \times MT} \tag{S21}$$

We next rewrite the ELBO to show the terms that depend on $K_j$. Let

$$L_n = \sum_{j=1}^p \left[ \frac{1}{2} \log |K_j^{-1}| - \frac{1}{2} \text{tr}(K_j^{-1} \langle \mathbf{x}_{n,j,:} \mathbf{x}_{n,j,:}^\top \rangle) \right] \tag{S22}$$

Then,

$$L(Q, \Omega) = \sum_{n=1}^N L_n + \text{const.} \tag{S23}$$

To optimize timescales, we first make the change of variables $\gamma_j = 1/\tau_j^2$. The variable $\gamma_j$ is simpler to work with. We then optimize with respect to $\gamma_j$. The $\gamma_j$ gradients are given by

$$\frac{\partial L}{\partial \gamma_j} = \sum_{n=1}^{N} \text{tr}\left(\left(\frac{\partial L_n}{\partial K_j}\right)^{\top}\left(\frac{\partial K_j}{\partial \gamma_j}\right)\right) \tag{S24}$$

where

$$\frac{\partial L_n}{\partial K_j} = -\frac{1}{2}K_j^{-1} + \frac{1}{2}K_j^{-1}\langle \mathbf{x}_{n,j,:}\mathbf{x}_{n,j,:}^{\top}\rangle K_j^{-1} \tag{S25}$$

and each element of $\partial K_j/\partial \gamma_j$ is given by

$$\frac{\partial k_{m_1,m_2,j}(t_1,t_2)}{\partial \gamma_j} = -\frac{1}{2}(\Delta t)^2\left(1 - \sigma_j^2\right)\exp\left(-\frac{1}{2}\gamma_j(\Delta t)^2\right) \tag{S26}$$

where $\Delta t$ is defined as in equation 9. To optimize $\gamma_j$ while respecting non-negativity constraints, we perform the change of variables $\gamma_j = \exp(\gamma_j^*)$, and then perform unconstrained gradient ascent with respect to $\gamma_j^*$.

Next, delay gradients for population $m$ and latent variable $j$ are given by

$$\frac{\partial L}{\partial D_{m,j}} = \sum_{n=1}^{N} \text{tr}\left(\left(\frac{\partial L_n}{\partial K_j}\right)^{\top}\left(\frac{\partial K_j}{\partial D_{m,j}}\right)\right) \tag{S27}$$

where $\frac{\partial L_n}{\partial K_j}$ is defined as in equation S25, and each element of $\partial K_j/\partial D_{m,j}$ is given by

$$\frac{\partial k_{m_1,m_2,j}(t_1,t_2)}{\partial D_{m,j}} = -\gamma_j(\Delta t)\left(1 - \sigma_j^2\right)\exp\left(-\frac{1}{2}\gamma_j(\Delta t)^2\right)\frac{\partial(\Delta t)}{\partial D_{m,j}} \tag{S28}$$

$$\frac{\partial(\Delta t)}{D_{m,j}} = \begin{cases} 1 & \text{if} \quad m = m_1 \\ -1 & \text{if} \quad m = m_2 \\ 0 & \text{otherwise} \end{cases} \tag{S29}$$

where $\Delta t$, $m_1$, and $m_2$ are defined as in equation 9. In practice, we fix all delay parameters for population 1 at 0 to ensure identifiability. Similar to the timescales, one might wish to constrain the delays within some physically realistic range, such as the length of an experimental trial, so that $-D_{\text{max}} \leq D_{m,j} \leq D_{\text{max}}$. Toward that end, we make the change of variables $D_{m,j} = D_{\text{max}} \cdot \tanh(\frac{D_{m,j}^*}{2})$ and perform unconstrained gradient ascent with respect to $D_{m,j}^*$. Here we chose $D_{\text{max}}$ to be half the length of a trial.

## S2.2 Evaluation of the lower bound

To monitor the progress of the fitting procedure, we evaluate the ELBO on each iteration. To evaluate the ELBO, we can rewrite it as follows:

$$L(Q,\Omega) = \mathbb{E}_Q[\log P(Y|\theta,\Omega)] - \text{KL}(Q(\theta)\|P(\theta|\Omega)) \tag{S30}$$

$\text{KL}(Q(\theta)\|P(\theta|\Omega))$ is the KL-divergence between the approximate posterior distribution $Q(\theta)$ and prior distribution $P(\theta|\Omega)$. Due to the factorized forms of $Q(\theta)$ and $P(\theta|\Omega)$, $L(Q,\Omega)$ becomes

$$L(Q,\Omega) = \mathbb{E}_Q[\log P(Y|\theta,\Omega)] - \text{KL}(Q_x(X)\|P(X|\Omega)) - \text{KL}(Q_c(C)\|P(C|\mathcal{A}))$$
$$- \text{KL}(Q_{\mathcal{A}}(\mathcal{A})\|P(\mathcal{A})) - \text{KL}(Q_\phi(\phi)\|P(\phi)) - \text{KL}(Q_d(\mathbf{d})\|P(\mathbf{d})) \tag{S31}$$

This form of the ELBO provides insight into the nature of the optimization procedure for fitting mDLAG models. The first term is the expected log-likelihood (with respect to the approximate posterior $Q(\theta)$) of the observed neural activity, $Y$, given the latest model parameters, $\theta$ and $\Omega$. This term encourages mDLAG models to explain the observed neural activity as well as possible. The KL-divergence terms, on the other hand, penalize deviations of each factor of the fitted posterior from its corresponding prior distribution, and hence act as a form of regularization.

Using the posterior updates in Section S2.1 and the prior definitions in Section 2, each term of the ELBO can be computed as follows:

$$\mathbb{E}_Q[\log P(Y|\theta,\Omega)] = -\frac{qNT}{2}\log(2\pi) + \frac{NT}{2}\sum_{m=1}^{M}\sum_{i=1}^{q_m}\langle\log\phi_i^m\rangle - \sum_{m=1}^{M}\sum_{i=1}^{q_m}(\widetilde{a}_\phi - \langle\phi_i^m\rangle b_\phi)$$

$$\text{(S32)}$$

$$-\text{KL}(Q_x(X)\|P(X|\Omega)) = \frac{MpNT}{2} + \frac{1}{2}\sum_{n=1}^{N}\left[\log|\bar{\Sigma}_x| - \sum_{j=1}^{p}\left[\log|K_j| + \text{tr}(K_j^{-1}\langle\mathbf{x}_{n,j,:}\mathbf{x}_{n,j,:}^\top\rangle)\right]\right]$$

$$\text{(S33)}$$

$$-\text{KL}(Q_c(C)\|P(C|\mathcal{A})) = \sum_{m=1}^{M}\left[\frac{q_m}{2}\sum_{j=1}^{p}\langle\log\alpha_j^m\rangle\right.$$

$$\left. + \frac{1}{2}\sum_{i=1}^{q_m}\left[\log|\Sigma_{c_i}^m| + \text{tr}(I_p - \langle\widetilde{\mathbf{c}}_i^m(\widetilde{\mathbf{c}}_i^m)^\top\rangle\langle\mathcal{A}^m\rangle)\right]\right]$$

$$\text{(S34)}$$

$$-\text{KL}(Q_{\mathcal{A}}(\mathcal{A})\|P(\mathcal{A})) = \sum_{m=1}^{M}\sum_{j=1}^{p}\left[-\widetilde{a}_\alpha^m\log\widetilde{b}_{\alpha,j}^m + a_\alpha\log b_\alpha + \log\frac{\Gamma(\widetilde{a}_\alpha^m)}{\Gamma(a_\alpha)} - b_\alpha\langle\alpha_j^m\rangle + \widetilde{a}_\alpha^m\right.$$

$$\left. + (a_\alpha - \widetilde{a}_\alpha^m)(\Psi(\widetilde{a}_\alpha^m) - \log\widetilde{b}_{\alpha,j}^m)\right]$$

$$\text{(S35)}$$

$$-\text{KL}(Q_\phi(\phi)\|P(\phi)) = \sum_{m=1}^{M}\sum_{i=1}^{q_m}\left[-\widetilde{a}_\phi\log\widetilde{b}_{\phi,i}^m + a_\phi\log b_\phi + \log\frac{\Gamma(\widetilde{a}_\phi)}{\Gamma(a_\phi)} - b_\phi\langle\phi_i^m\rangle + \widetilde{a}_\phi\right.$$

$$\left. + (a_\phi - \widetilde{a}_\phi)(\Psi(\widetilde{a}_\phi) - \log\widetilde{b}_{\phi,i}^m)\right]$$

$$\text{(S36)}$$

$$-\text{KL}(Q_d(\mathbf{d})\|P(\mathbf{d})) = \frac{q}{2} + \frac{q}{2}\log\beta + \frac{1}{2}\log|\Sigma_d| - \frac{1}{2}\beta\langle\|\mathbf{d}\|_2^2\rangle$$

$$\text{(S37)}$$

Here, $\Gamma(\cdot)$ is the gamma function, and $\Psi(\cdot)$ is the digamma function. All moments $\langle\cdot\rangle$ can be readily computed from the approximate posterior distributions given in equations S6–S10.

### S2.3 Parameter initialization and removal of insignificant latent variables

To initialize the mDLAG fitting procedure, we first specified an initial number of latent variables, $p$. Through automatic relevance determination, mDLAG effectively prunes insignificant latent variables. We leveraged this feature to improve the computational efficiency (with respect to both speed and memory) of the fitting procedure as follows. Each iteration, we evaluated the sample second moment of the estimated latent variables, $\frac{1}{N}\sum_n\bar{\mu}_{x_n}^2$ (here, $\bar{\mu}_{x_n}^2$ is computed by squaring each element of $\bar{\mu}_{x_n}$). If the sample second moment of a latent variable was not larger than some threshold, $\epsilon$, for at least one population, then we removed it from the mDLAG model (and its associated parameters in $\theta$ and $\Omega$) [4]. Here, we chose $\epsilon = 10^{-7}$. We chose an initial $p$ to be as small as possible (to minimize runtime) yet large enough that at least one of the initial latent variables would be deemed insignificant (according to the criterion above or according to shared variance explained, see equation S52), thus ensuring that dimensionalities were not underestimated. A value of $p = 30$ was sufficient for the datasets analyzed here.

To initialize the rest of the mDLAG fitting procedure, we specified initial values for only the moments of the posterior factors $Q_d(\mathbf{d})$, $Q_\phi(\phi)$, $Q_c(C)$, and $Q_{\mathcal{A}}(\mathcal{A})$ (equations S7–S10) that were needed to begin iteration. $Q_x(X)$ was then the first factor to be updated each iteration of the fitting procedure. We specified noninformative priors by fixing all hyperparameters to a very small value [4], $\beta, a_\phi, b_\phi, a_\alpha, b_\alpha = 10^{-12}$. For $Q_d(\mathbf{d})$, we initialized $\mu_d^m$ at the sample mean of neural activity across all trials and time points. For $Q_\phi(\phi)$, we initialized $\langle\phi_i^m\rangle^{-1}$ for each neuron $i$ in population $m$ to the sample variance of that neuron across all trials and time points. For $Q_c(C)$, we first randomly initialized all first moments $\widetilde{\mu}_{c_i}^m$ with entries drawn from a zero-mean Gaussian distribution with variance chosen to match the scale of the data. Then, we initialized the second moments $\langle\widetilde{\mathbf{c}}_i^m(\widetilde{\mathbf{c}}_i^m)^\top\rangle$ to the outer product of first moments $\widetilde{\mu}_{c_i}^m\widetilde{\mu}_{c_i}^{m\top}$. For $Q_{\mathcal{A}}(\mathcal{A})$, we initialized $\langle\alpha_j^m\rangle$ for each latent $j$ in

population $m$ to $q_m/\langle\|\mathbf{c}_j^m\|_2^2\rangle$, which stems from equations S19 and S20. Finally, we initialized all delays to zero, and all Gaussian process timescales to the same value, equal to twice the sampling period or spike count bin width of the neural activity.

## S3 Group factor analysis (GFA) with anisotropic observation noise

Throughout this work, we included the static dimensionality reduction method group factor analysis (GFA) as a comparison benchmark for mDLAG (Fig. 4, Supplementary Fig. S2b, Supplementary Fig. S3, Supplementary Fig. S4). We implemented the following modified version of GFA with anisotropic observation noise (contrasting the isotropic model in [4], see below). For population $m$ on trial $n$, define a linear relationship between observed neural activity, $\mathbf{y}_n^m \in \mathbb{R}^{q_m}$, and latent variables, $\mathbf{x}_n \in \mathbb{R}^p$:

$$\mathbf{y}_n^m = C^m\mathbf{x}_n + \mathbf{d}^m + \boldsymbol{\varepsilon}^m \tag{S38}$$

$$\boldsymbol{\varepsilon}^m \sim \mathcal{N}(\mathbf{0}, (\Phi^m)^{-1}) \tag{S39}$$

where $C^m \in \mathbb{R}^{q_m \times p}$, $\mathbf{d}^m \in \mathbb{R}^{q_m}$, and $\Phi^m \in \mathbb{S}^{q_m \times q_m}$ are probabilistic model parameters with prior distributions, defined below.

The parameter $\mathbf{d}^m$ can be thought of as the mean firing rate of each neuron in population $m$. Each $\mathbf{d}^m$ is defined to have a Gaussian prior:

$$P(\mathbf{d}^m) = \mathcal{N}(\mathbf{d}^m \mid \mathbf{0}, \beta^{-1}I_{q_m}) \tag{S40}$$

where $\beta \in \mathbb{R}_{>0}$ is a hyperparameter, and $I_{q_m}$ is the $q_m \times q_m$ identity matrix. Here we constrain the precision matrix $\Phi^m = \mathrm{diag}(\phi_1^m, \ldots, \phi_{q_m}^m)$ to be diagonal to capture variance that is independent to each neuron (in [4], the precision matrix is defined as $\tau I_{q_m}$, so that the noise variance, $\tau^{-1} \in \mathbb{R}_{>0}$, is the same for all neurons). This constraint encourages the latent variables to explain as much of the shared variance among neurons as possible. We set the conjugate Gamma prior over each $\phi_i^m$, for each neuron $i = 1, \ldots, q_m$:

$$P(\phi_i^m) = \Gamma(\phi_i^m \mid a_\phi, b_\phi) \tag{S41}$$

where $a_\phi, b_\phi \in \mathbb{R}_{>0}$ are hyperparameters.

The loading matrix $C^m$ linearly combines latent variables and maps them to observed neural activity. The automatic selection of the number of latents, and of the number of populations a particular latent involves, is accomplished through automatic relevance determination (ARD; see also [3]). Specifically, each column of $C^m$ is defined by the following prior:

$$P(\mathbf{c}_j^m \mid \alpha_j^m) = \mathcal{N}(\mathbf{c}_j^m \mid \mathbf{0}, (\alpha_j^m)^{-1}I_{q_m}) \tag{S42}$$

$$P(\alpha_j^m) = \Gamma(\alpha_j^m \mid a_\alpha, b_\alpha) \tag{S43}$$

where $\mathbf{c}_j^m \in \mathbb{R}^{q_m}$ is the $j^{\text{th}}$ column of $C^m$, $\alpha_j^m \in \mathbb{R}_{>0}$ is the ARD parameter for latent $j$ and population $m$, and $a_\alpha, b_\alpha \in \mathbb{R}_{>0}$ are hyperparameters. The ARD prior encourages population-wise sparsity for each latent variable.

Finally, latent variables $\mathbf{x}_n$ are defined by a standard Normal prior:

$$P(\mathbf{x}_n) = \mathcal{N}(\mathbf{x}_n \mid \mathbf{0}, I_p) \tag{S44}$$

where $I_p$ is the $p \times p$ identity matrix. As with mDLAG, GFA models are fit using approximate inference: posterior estimates maximize a variational lower bound on the data likelihood, and are constrained to follow a factorized form (see Section S2).

## S4 Leave-group-out prediction

To facilitate comparison of performance across methods (Supplementary Fig. S2, Supplementary Fig. S3), we developed a leave-group-out prediction procedure that measures an mDLAG model's ability to capture interactions across populations [2, 4]. In brief, we used a model fit to training data to predict (on held-out test trials) the unobserved activity of held-out neurons in one population, given the observed activity of neurons in the remaining populations.

In detail, let us first collect observed variables (for one trial) in a manner that highlights group structure. We collect observations in population $m$ on trial $n$ in $\widetilde{\mathbf{y}}_n^m = [\mathbf{y}_{n,:,1}^{m\top} \cdots \mathbf{y}_{n,:,T}^{m\top}]^\top \in \mathbb{R}^{q_m T}$ by vertically concatenating the observed neural activity $\mathbf{y}_{n,:,t}^m \in \mathbb{R}^{q_m}$ in population $m$ across all times $t = 1, \ldots, T$. Then, we collect the observations for the remaining $M - 1$ populations in $\widetilde{\mathbf{y}}_n^{-m} \in \mathbb{R}^{\sum_{\ell \neq m} q_\ell T}$, obtained by vertically concatenating the ordered set of observations $\{\widetilde{\mathbf{y}}_n^\ell\}_{\ell \neq m}$.

Our goal is to predict $\widetilde{\mathbf{y}}_n^m$ given $\widetilde{\mathbf{y}}_n^{-m}$. We do so by inferring the latent variables given observations $\widetilde{\mathbf{y}}_n^{-m}$, and then predicting the held-out activity $\widetilde{\mathbf{y}}_n^m$ from the inferred latent variables. Toward that end, we similarly collect the latent variables for population $m$ on trial $n$ in $\widetilde{\mathbf{x}}_n^m = [\mathbf{x}_{n,:,1}^{m\top} \cdots \mathbf{x}_{n,:,T}^{m\top}]^\top \in \mathbb{R}^{pT}$ by vertically concatenating the latent variables $\mathbf{x}_{n,:,t}^m \in \mathbb{R}^p$ across all times $t = 1, \ldots, T$. Then the latent variables for the remaining $M - 1$ populations can be collected in $\widetilde{\mathbf{x}}_n^{-m} \in \mathbb{R}^{(M-1)pT}$, obtained by vertically concatenating the ordered set of latent variables $\{\widetilde{\mathbf{x}}_n^\ell\}_{\ell \neq m}$. This variable reorganization then allows us to rewrite the mDLAG state model as

$$\begin{bmatrix} \widetilde{\mathbf{x}}_n^m \\ \widetilde{\mathbf{x}}_n^{-m} \end{bmatrix} \sim \mathcal{N}\left(\mathbf{0}, \begin{bmatrix} \widetilde{K}_{m,m} & \widetilde{K}_{m,-m} \\ \widetilde{K}_{-m,m} & \widetilde{K}_{-m,-m} \end{bmatrix}\right) \tag{S45}$$

where the elements of the GP covariance matrices $\widetilde{K}_{m,m} \in \mathbb{S}^{pT \times pT}$, $\widetilde{K}_{m,-m} = \widetilde{K}_{-m,m}^\top \in \mathbb{R}^{pT \times (M-1)pT}$, and $\widetilde{K}_{-m,-m} \in \mathbb{S}^{(M-1)pT \times (M-1)pT}$ are computed using equations 8 and 9.

Next, for each population $m$, define $\langle\widetilde{C}^m\rangle \in \mathbb{R}^{q_m T \times pT}$, $\langle\widetilde{\Phi}^m\rangle \in \mathbb{S}^{q_m T \times q_m T}$, and $\langle\widetilde{W}^m\rangle \in \mathbb{R}^{pT \times pT}$ as block diagonal matrices comprising $T$ copies of the matrices $\langle C^m\rangle$, $\langle\Phi^m\rangle$, and $\langle W^m\rangle = \langle C^{m\top}\Phi^m C^m\rangle$, respectively. Define also $\langle\widetilde{\mathbf{d}}^m\rangle \in \mathbb{R}^{q_m T}$ by vertically concatenating $T$ copies of $\langle\mathbf{d}^m\rangle$. The parameters corresponding to the remaining $M - 1$ populations can then be collected into the block diagonal matrices $\langle\widetilde{C}^{-m}\rangle = \text{diag}(\{\langle\widetilde{C}^\ell\rangle\}_{\ell \neq m}) \in \mathbb{R}^{\sum_{\ell \neq m} q_\ell T \times (M-1)pT}$, $\langle\widetilde{\Phi}^{-m}\rangle = \text{diag}(\{\langle\widetilde{\Phi}^\ell\rangle\}_{\ell \neq m}) \in \mathbb{R}^{\sum_{\ell \neq m} q_\ell T \times \sum_{\ell \neq m} q_\ell T}$, $\langle\widetilde{W}^{-m}\rangle = \text{diag}(\{\langle\widetilde{W}^\ell\rangle\}_{\ell \neq m}) \in \mathbb{R}^{(M-1)pT \times (M-1)pT}$, and the vector $\langle\widetilde{\mathbf{d}}^{-m}\rangle \in \mathbb{R}^{\sum_{\ell \neq m} q_\ell T}$, obtained by vertically concatenating the elements of the set $\{\langle\widetilde{\mathbf{d}}^\ell\rangle\}_{\ell \neq m}$.

Similar to the updates given by equations S11 and S12, we compute the inferred latent variables given the observed neural activity $\widetilde{\mathbf{y}}_n^{-m}$ according to

$$\widetilde{\Sigma}_x^{-m} = \left(\widetilde{K}_{-m,-m}^{-1} + \langle\widetilde{W}^{-m}\rangle\right)^{-1} \tag{S46}$$

$$\widetilde{\boldsymbol{\mu}}_{x_n}^{-m} = \widetilde{\Sigma}_x^{-m} \langle\widetilde{C}^{-m}\rangle^\top \langle\widetilde{\Phi}^{-m}\rangle \left(\widetilde{\mathbf{y}}_n^{-m} - \langle\widetilde{\mathbf{d}}^{-m}\rangle\right) \tag{S47}$$

We then use equation S45 to infer the latent variables for population $m$ according to

$$\widetilde{\boldsymbol{\mu}}_{x_n}^m = \widetilde{K}_{m,-m}(\widetilde{K}_{-m})^{-1}\widetilde{\boldsymbol{\mu}}_{x_n}^{-m} \tag{S48}$$

and take predictions of the neural activity in population $m$ to be

$$\hat{\widetilde{\mathbf{y}}}_n^m = \langle\widetilde{C}^m\rangle\widetilde{\boldsymbol{\mu}}_{x_n}^m + \langle\widetilde{\mathbf{d}}^m\rangle \tag{S49}$$

We next use equation S49 to define a measure of a model's across-group predictive performance. Assume we are given an mDLAG model fit to training data. Then let $\widetilde{\mathbf{y}}_n^m$ be the activity of population $m$ on trial $n$ of a held-out test set, and let $\hat{\widetilde{\mathbf{y}}}_n^m$ be its predicted value given by equation S49. Collect these values across all $n = 1, \ldots, N$ held-out test set trials into the respective matrices $Y^m = [\widetilde{\mathbf{y}}_1^m \cdots \widetilde{\mathbf{y}}_N^m] \in \mathbb{R}^{q_m T \times N}$ and $\hat{Y}^m = [\hat{\widetilde{\mathbf{y}}}_1^m \cdots \hat{\widetilde{\mathbf{y}}}_N^m] \in \mathbb{R}^{q_m T \times N}$. Furthermore, let $\boldsymbol{\mu}_y^m \in \mathbb{R}^{q_m}$ be the sample mean for each neuron in population $m$, taken over all time points and trials. Construct $\bar{\mathbf{y}}^m = [\boldsymbol{\mu}_y^{m\top} \cdots \boldsymbol{\mu}_y^{m\top}]^\top \in \mathbb{R}^{q_m T}$ by vertically concatenating $T$ copies of the sample mean, and construct $\bar{Y}^m = [\bar{\mathbf{y}}^m \cdots \bar{\mathbf{y}}^m] \in \mathbb{R}^{q_m T \times N}$ by horizontally concatenating $N$ copies of $\bar{\mathbf{y}}^m$.

We then define a leave-group-out $R^2$ value as follows:

$$R_{\text{lgo}}^2 = 1 - \frac{\sum_{m=1}^M \|Y^m - \hat{Y}^m\|_F^2}{\sum_{m=1}^M \|Y^m - \bar{Y}^m\|_F^2} \tag{S50}$$

$R_{lgo}^2 \in (-\infty, 1]$, where a value of 1 implies perfect prediction of neural activity, and a negative value implies that estimates predict neural activity less accurately than simply the sample mean. $R_{\text{lgo}}^2$ is normalized by the total variance of neural activity within each dataset, thereby facilitating comparison across datasets, in which the variance of neural activity could vary widely.

## S5    Choosing the number of significant latent variables in each population

**mDLAG**    mDLAG incorporates ARD to automatically determine, during model fitting, both the total number of latent variables and the subset of populations that each latent involves. We sought an intuitive measure of the significance of each latent variable within a population, post-fitting, based on the amount of shared variance each latent variable explains. The shared variance of latent variable $j$ in population $m$ is given by $\langle \|\mathbf{c}_j^m\|_2^2 \rangle$, the expected squared magnitude of the $j^{\text{th}}$ column of the loading matrix $C^m$. Since the total shared variance can vary widely across populations, we considered a normalized metric, the fraction of shared variance explained by latent variable $j$ in population $m$ (displayed in Supplementary Fig. S1b):

$$\nu_j^m = \frac{\langle \|\mathbf{c}_j^m\|_2^2 \rangle}{\sum_{k=1}^p \langle \|\mathbf{c}_k^m\|_2^2 \rangle} \tag{S51}$$

For small ARD hyperparameters, $a_\alpha$ and $b_\alpha$ (as we have chosen in this work), the fraction of shared variance can equivalently be computed using the estimated ARD parameters (see equations S19 and S20):

$$\nu_j^m \approx \frac{\langle \alpha_j^m \rangle^{-1}}{\sum_{k=1}^p \langle \alpha_k^m \rangle^{-1}} \tag{S52}$$

If a latent variable does not significantly explain activity in a population, then $\nu_j^m$ will be close to zero (see Supplementary Fig. S1b, 'V1' and 'V2'). In our analysis of the Neuropixels recordings (Section 4, Fig. 5) we reported a latent variable as significant in a population if it explained at least 2% of the shared variance within that population ($\nu_j^m \geq 0.02$).

**DLAG**    For comparison on the V1-V2 recordings (Supplementary Fig. S1d, Supplementary Fig. S2a), we also applied DLAG [5], which was designed to study concurrent signaling between two populations. In contrast with mDLAG, DLAG explicitly designates latent variables as "across-population" and "within-population." Then the number of each type of latent variable (resulting in three hyperparameters) is determined via a heuristic two-stage cross-validation procedure (used in [2]): (1) Factor analysis (FA) models [6] are applied to each population separately, and cross-validation is used to identify the total dimensionality of each population. (2) A reduced space of candidate DLAG models are considered, such that the number of each population's within- and across-population latent variables sum to that populations's optimal FA dimensionality. From this reduced space of model candidates, the optimal DLAG model is identified using cross-validation. We employed this approach here on the training trials of the V1-V2 datasets, using 4-fold cross-validation in both stages.

# S6 Supplementary Figures

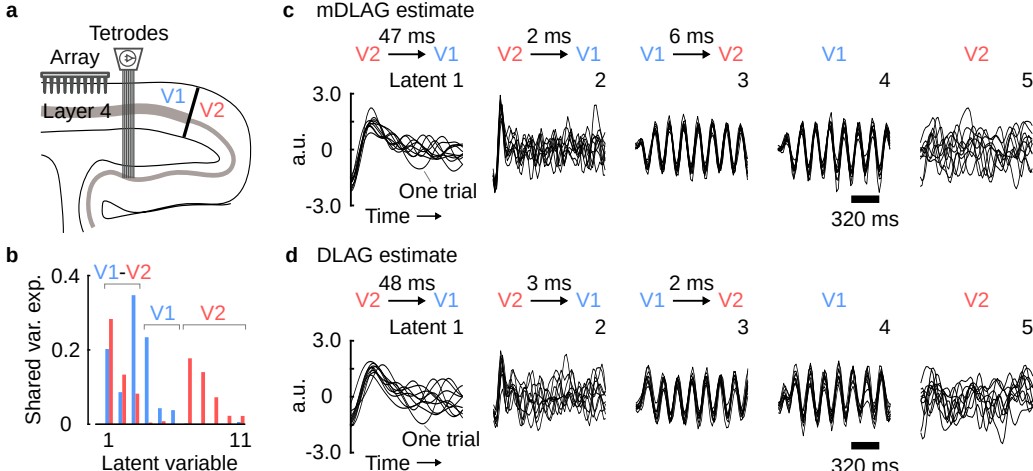

Figure S1: mDLAG detects hallmarks of V1-V2 communication. (**a**) Schematic showing a sagittal section of occipital cortex and the recording setup. V1 population activity was recorded using a 96-channel Utah array. V2 population activity was recorded using a set of movable electrodes and tetrodes. (**b**) Fraction of shared variance explained ('var. exp.', see equation S52) within each population by mDLAG latent variables for a representative dataset. Latent variables are grouped from left to right based on whether they are shared between V1 and V2 ('V1-V2'), local to V1 ('V1'), or local to V2 ('V2'). Only latent variables that explained at least 2% of the shared variance in at least one area are shown. (**c**) mDLAG single-trial latent time course estimates for a representative dataset (same as in (b)). Only the top local latent variable is shown for each population ('V1' and 'V2'). Same conventions as in Fig. 4b. (**d**) DLAG single-trial latent time course estimates for the same dataset as in (c). Same conventions as in Fig. 4b.

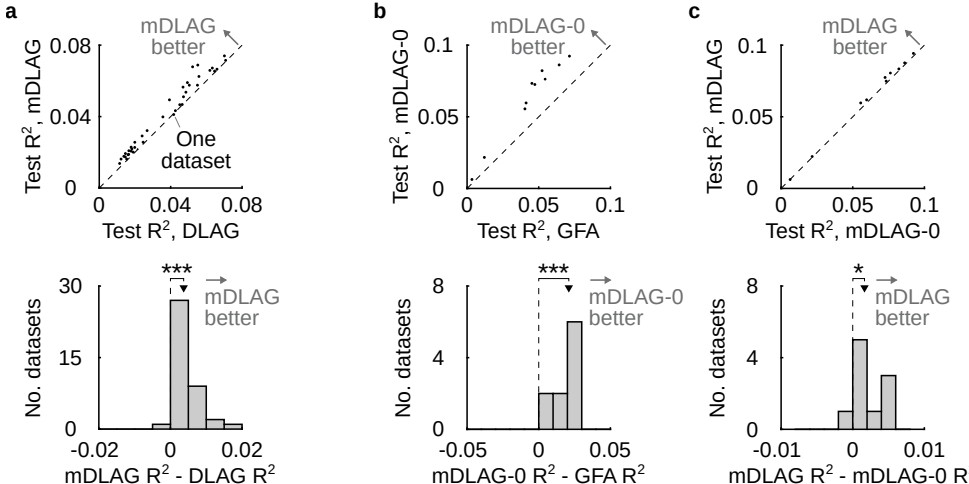

Figure S2: mDLAG performance across neural recordings. (**a**) mDLAG outperforms DLAG across V1-V2 datasets (related to Supplementary Fig. S1). Here we consider two neuronal populations. For three or more populations, DLAG cannot be applied directly. Top: mDLAG performance versus DLAG performance (leave-group-out $R^2$ evaluated on 100 test trials). Each data point represents one V1-V2 dataset. Bottom: Distribution of differences in performance between mDLAG and DLAG. mDLAG significantly outperformed DLAG across datasets ($\star\star\star$: one-sided paired sign test; $p = 3.7 \times 10^{-11}$), indicating the performance benefit of ARD for model selection. (**b**)-(**c**) Demonstrating the empirical benefit of mDLAG's use of Gaussian processes and time delays on the Neuropixels recordings (related to Fig. 5). Here we consider three neuronal populations. (**b**) mDLAG models for which time delays were fixed at zero ('mDLAG-0') outperform GFA across Neuropixels datasets. Top: mDLAG-0 performance versus GFA performance (leave-group-out $R^2$ evaluated on 75 test trials). Each data point represents one Neuropixels dataset. Bottom: Distribution of differences in performance between mDLAG-0 and GFA. mDLAG-0 significantly outperformed GFA across datasets ($\star\star\star$: one-sided paired sign test; $p = 9.8 \times 10^{-4}$), indicating the performance benefit of including a Gaussian process time series model. (**c**) mDLAG models with estimated time delays outperform mDLAG models with time delays fixed at zero ('mDLAG-0') across Neuropixels datasets. Top: mDLAG performance versus mDLAG-0 performance (leave-group-out $R^2$ evaluated on 75 test trials). Each data point represents one Neuropixels dataset. Bottom: Distribution of differences in performance between mDLAG and mDLAG-0. mDLAG significantly outperformed mDLAG-0 across datasets ($\star$: one-sided paired sign test; $p = 0.0107$), indicating the performance benefit of estimating time delays between populations.

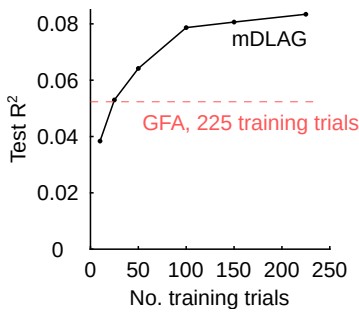

Figure S3: mDLAG test performance (leave-group-out $R^2$ evaluated on 75 held-out trials) versus the number of available training trials. We re-fit mDLAG models to the Neuropixels dataset analyzed in Fig. 5, but we limited the number of experimental trials available in the training set (i.e., we fit mDLAG to training sets with 10, 25, 50, 100, 150, up to 225 trials—the full training set size). With as few as 100 training trials (less than half of the full training set size), mDLAG's test performance achieved nearly 95% of full performance (black curve). Furthermore, with as few as 25 training trials, mDLAG still outperformed the group factor analysis (GFA) model fit to all 225 training trials (red dashed line). This example demonstrates empirically that mDLAG performs well with trial counts typical of neurophysiological experiments. In general, this data-efficiency is due to mDLAG's incorporation of (1) dimensionality reduction, (2) temporal smoothing, and (3) automatic relevance determination. These components act as forms of regularization that benefit model performance in data-limited regimes.

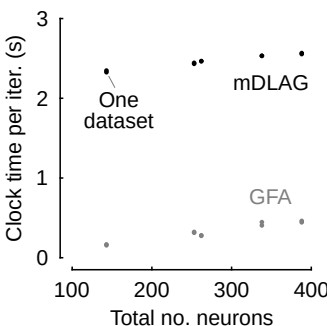

Figure S4: mDLAG (black) and GFA (gray) runtimes on the Neuropixels recordings. Each point corresponds to one dataset: 10 datasets are shown for each method (some points overlap one another; same datasets as in Supplementary Fig. S2b,c; $M = 3$ populations, $N = 225$ training trials, and $T = 50$ time points per trial). The average clock time per fitting iteration (in seconds) for either method scales linearly as a function of total number of neurons analyzed. For mDLAG, increasing the number of analyzed neurons from 143 to 388 (an increase by a factor of 2.7) resulted in a 10% increase in runtime. Overall, mDLAG is more computationally intensive than GFA due to the incorporation of Gaussian processes. Each mDLAG fitting iteration requires the inversion of a $MpT \times MpT$ matrix (equation S11). We conservatively ran each mDLAG model for 50,000 iterations, resulting in an average total runtime of 34 hours. Had we used a less conservative, but still reasonable, stopping tolerance of $10^{-6}$, then the average number of iterations required for convergence would have been 17,000 (with similar parameter estimates), for an average total runtime of 11.5 hours. Results were obtained on a Red Hat Enterprise Linux machine (release 7.9, 64-bit) with 250GB of RAM running Matlab (R2019a), on an Intel Xeon CPU (E5-2695 v3, 2.3 GHz).

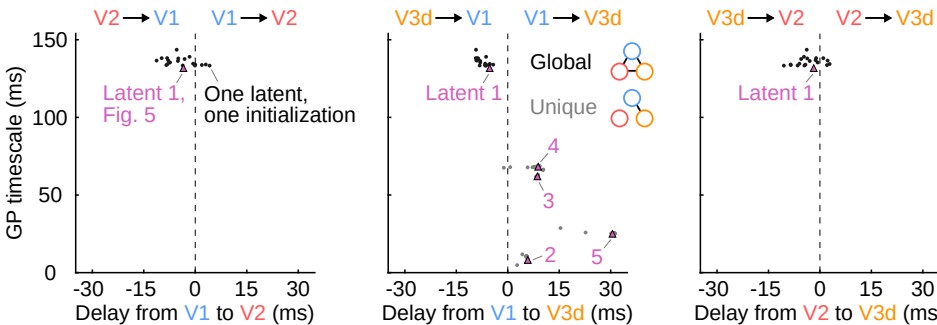

Figure S5: Sensitivity of results (Fig. 5) to mDLAG initialization. To investigate the sensitivity of our results (Fig. 5) to the initialization of the mDLAG fitting procedure (see Supplementary Section S2.3), we ran the analysis of the example dataset 20 times, starting from a different random initialization each time. Estimated dimensionalities (Fig. 5e) were identical across all 20 runs. Leave-group-out predictive performance (Supplementary Fig. S2c) was also highly consistent (test $R^2$ for example run shown in Fig. 5: 0.0834; best $R^2$: 0.0842; worst $R^2$: 0.0834). Above, we show the distribution of Gaussian process (GP) timescales and time delays across the 20 runs. From left to right: GP timescale versus relative delay from V1 to V2, from V1 to V3d, and from V2 to V3d, respectively. Each point represents one latent variable. Black points ("global") correspond to the dimension shared globally across all three populations. Gray points ("unique") correspond to the dimensions unique to V1 and V3d. Magenta triangles correspond to the latent variables estimated for the example run shown in Fig. 5 (the time courses of latents 1–3 are displayed in Fig. 5f). Clustered around each magenta triangle are 20 points, which may occlude one another due to their proximity. These analyses suggest that the results in Fig. 5 are largely insensitive to the initialization of the mDLAG fitting prodedure.