# OpenReview forum: "Uncovering motifs of concurrent signaling across multiple neuronal populations"
_NeurIPS.cc/2023/Conference — NeurIPS 2023 spotlight_

### Official Review · Reviewer_EMi6 · 2023-07-01

**Soundness:** 3 good
**Presentation:** 4 excellent
**Contribution:** 3 good
**Rating:** 7
**Confidence:** 4

**Summary:**

This work extends an established line of previously proposed methods for Gaussian Process factor analysis-based models of neuroscience data. In comparison with previously published methods, it extends the factor model to include correlations and time delays between different (matched) latents underlying different populations. Inference is performed via mostly closed-form variational updates. This method is applied to data from paired recordings from three regions in visual cortex, providing evidence of directional influence between them in the context of overlapping receptive fields.

This is a valuable contribution that extends previous methods in ways that are likely to prove increasingly useful in an era of widespread multi-region recordings. While the algorithmic innovation is somewhat incremental, the validation, presentation, and contextualization to questions in neuroscience are very good.

**Strengths:**

- This is a valuable contribution that extends previous methods in ways that are likely to prove increasingly useful in an era of widespread multi-region recordings. While the algorithmic innovation is somewhat incremental, the validation, presentation, and contextualization to questions in neuroscience are very good.
- The manuscript is clearly and carefully written, including a helpful notation key in the supplement.
- Rigorous model evaluation through recovery of parameters on synthetic data and careful analysis of a challenge data set.
- Details of the model and its inference are carefully laid out in the main text and supplement.

**Weaknesses:**

- While the authors have done a commendable job of using a consistent and careful notation, the resulting formulas are index-dense, and it may be easy for some readers to miss the key conceptual setup, particularly which latents are independent of which other latents. For example, l. 94 in the supplement clearly notes that latent posterior estimates are independent for different trials, and this is implicit in (7), but this probably deserves a mention in the main text. Similarly, different $\mathbf{x}_j$ for $j=1\ldots p$ appear to also be independent, and this is a substantive assumption (see comment under limitations below), but I didn't see this discussed (though I may have missed it).

**Questions:**

- The authors report several latents with nearly sinusoidal form. Are inferred delays in these cases well-defined? Is it possible to infer directionality in these cases, when a lag of $2\pi(1  - D/\tau)$ (with $\tau$ the period) should also produce similar results? This seems to me clearer in cases of sparse signals or strong transients but more difficult when latents are densely active.

- While the authors cite a number of related factor analysis models in neuroscience (especially refs 14, 15), there is a limited discussion of some other lines of multi-region work (e.g., ref 33, but also Gallagher et al. NeurIPS 2017 from the same group) that have previously used a GP-based factor analysis model and included relative delays between regions. What, precisely, is the difference between these models apart from their application to spikes as opposed to LFP?

**Limitations:**

- From (7), it appears as if the latents for each $j=1\ldots p$ are independent: $\mathrm{cov}[x^m_{njt}x^m_{nj't}] \propto \delta_{jj'}$. That is, there is potential coupling _across populations_ for each latent but not _across latents_ even within a population. This choice might warrant some additional discussion in the text.
- The variational inference algorithm for the latents requires inverting an $MpT \times MpT$ matrix for each trial. I suspect this is feasible for typical numbers ($M \sim 3$, $p \sim 10$, $T \sim 10$) and can be parallelized across trials, but will scale poorly if any of the relevant parameters become large.
- While the model is fit to spike count data, the observation model in (1) is Gaussian. This is likely to hinder inference when bins are small and/or counts are low.

---

> ### Author Rebuttal · Authors · 2023-08-10
>
> > **Weaknesses**
>
> Thanks to the reviewer, we realize we could have been clearer about the independence structure of the latents *a priori* and *a posteriori*. We have added the following text:
> - Section 2: "Under equation 7, latents are independent and identically distributed across trials."
> - Supp. Section S2.1.1: "Note that, under the posterior distribution, latent variables $j = 1,\ldots,p$ are no longer independent, as they are under the prior distribution (equations 7, S2)."
>
> We hope that these statements better complement Supp. Section S2.1.1, in which we preface each posterior update (eqs. S11-S20) with a statement about independence structure. Please see also our response under Limitations, below.
>
> > **Questions**
>
> > - Sinusoids
>
> The reviewer's intuition regarding purely sinusoidal signals is correct. The central quantity is the latent's underlying covariance function. For example, a signal generated from a squared exponential function (Fig. 2c) would be "densely active," but the correct time delay can be unambiguously identified via the peak of the cross-covariance function. A sinusoidal signal has a sinusoidal covariance function, which has translational symmetry (i.e., it is periodic). Hence two time delay estimates spaced one period apart present two local optima.
>
> This potential ambiguity can be resolved by additional context. For example, latent 3 of Fig. 5f is nearly sinusoidal with period 333 ms (corresponding to the grating stimulus) and estimated time delay +10 ms. An initial transient, however, breaks its translational symmetry (the 1st cycle is different from the 2nd and 3rd cycle). Furthermore, potential alternative delay estimates, -323 ms and +343 ms, would be inconsistent with the response latencies of the neurons in V1 and V3d (Fig. 5b; both populations respond within 10s of ms of stimulus onset). We can thus conclude that +10 ms is a reasonable estimate consistent with additional observations. Finally, we note that the sinusoidal signals identified here are due to experimental design: future experiments could be designed to avoid them.
>
> > - Related models
>
> We thank the reviewer for this reference, and now cite it. In brief, we see three axes that distinguish mDLAG from the cross-spectral factor analysis (CSFA) approach of Gallagher et al., 2017: (1) group structure, (2) parametrization of time delays, and (3) automatic relevance determination (ARD). The design choices along these axes give mDLAG interpretational and computational advantages in the study of multi-population recordings.
>
> In detail:
> 1. Group structure: While Gallagher et al. indeed analyze activity in different brain regions, each "brain region" corresponds to a single LFP channel or time series. Consequently, there is no explicit group structure (i.e., the capability that brain regions may be multi-variate) built into CSFA, either in the observation model (eqs. 4 and 5 of Gallagher et al.) or the state model (eqs. 1 and 6 of Gallagher et al.). CSFA appears to be similar to time-delay GPFA (Lakshmanan et al., 2015), which studied time delays between pairs of individual neurons. In contrast, mDLAG incorporates group structure into both the observation (eqs. 1-6) and state models (eqs. 7-9; time delays are shared across neurons in the same population). Group structure facilitates inferences about network-level interactions (Fig. 1c, Fig. 2b) and about signal flow between populations (Fig. 1b), rather than between pairs of individual neurons or LFP channels.
>
> 2. Parametrization of time delays: Gallagher et al. use the cross-spectral mixture kernel (Ulrich et al., 2015). Accordingly, (1) the GP covariance function of each latent variable is a weighted sum of spectral Gaussian kernels, and (2) for each latent variable and between each pair of LFP channels, each spectral Gaussian component contributes a constant phase difference at its center frequency. Effectively, "time delays" between a pair of LFP channels are parametrized as a piecewise-constant phase function in the frequency domain. In contrast, (1) the GP covariance function of each mDLAG latent variable comprises one component (e.g., the squared exponential function), and (2) for each latent variable and between each pair of populations, mDLAG directly defines a time delay, or equivalently, a linear (as opposed to piecewise-constant) phase function in the frequency domain. These design choices enable a simpler description of cross-population temporal structure (fewer parameters) while remaining reasonably expressive.
>
> 3. ARD: CSFA requires cross-validation over a space of four hyperparameters. mDLAG employs ARD to enable computationally tractable model selection for multi-population recordings.
>
> > **Limitations**
>
> > - Independence
>
> Indeed, under the prior distribution, eq. 7 (see also Supp. eq. S2), the latents $x_{n,j,:}$, $j =1,\ldots,p$ are independent. Under the posterior distribution, however, this independence no longer holds (see Supp. eqs. S6, S11, and S12). The interaction between the structure of the prior and posterior distributions can be understood by inspection of the ELBO (Supp. Section S2.2, eq. S31). The KL-divergence term $KL(Q_x(X) || P(X|\Omega))$ penalizes deviations of the posterior distribution $Q_x(X)$ from the prior distribution $P(X|\Omega)$. The independence structure of the prior distribution therefore acts as a form of regularization, not a hard constraint.
>
> > - Variational inference
>
> We thank the reviewer for the opportunity to clarify an important point: the latent posterior covariance (Supp. eq. S11) is identical for all trials of the same length. This computation can therefore be reused efficiently across trials. We have added this statement to Supp. Section S2.1.1. Please see also the General Response to Reviewers, "Computational demands."
>
> > - Gaussian model
>
> We thank the reviewer for making this point. Please see the General Response to Reviewers, "Gaussian observation model applied to spike counts."

---

> > ### Comment · Reviewer_EMi6 · 2023-08-10
> >
> > I appreciate the authors' responses to my and the other reviewers' critiques. I believe this is a valuable method that will likely find application in to those doing multi-region recordings in neuroscience.

---

### Official Review · Reviewer_qoeh · 2023-07-03

**Soundness:** 4 excellent
**Presentation:** 4 excellent
**Contribution:** 4 excellent
**Rating:** 8
**Confidence:** 4

**Summary:**

This paper develops a new approach to analyze multi-region neural populations recordings. This approach extends DLAG to analyze communication across more than 2 regions. The main technical novelty lies in tractably extending the model definition of DLAG to multiple regions through the incorporation of the ARD prior. I found the approach elegant and particularly relevant to the current state of neuroscience research where multi-region recordings are growing in number. DLAG and mDLAG like approaches offer the ability to temporally disentangle concurrent signals, allowing us to make sense of multi-region recordings. The experiments are thorough and all the figures are very well made and described. Overall, I very much enjoyed reading this!

**Strengths:**

1. The paper is extremely relevant for a computational neuroscience crowd, where analyzing multi-region recordings is an important challenge. This approach can in practice help drive neuroscience research and answer questions about signaling across regions in the brain.
2. Technically, the novelty lies in the model description (which allows for easy disentangling of latents involved across region(s)) and in discovering a tractable inference scheme (I should say that I did not read details of the inference in the supplement).
3. The simulations and real-world experiments are both very thorough and nicely done. Details in the supplement were also helpful in contextualizing some of the results.
4. The paper is very well-written, and the figures are very clean!

**Weaknesses:**

 1. The main question that I found myself struggling with was the order of latent variables across regions. I realize that there in a GP connecting them, but that does not guarantee that the ordering of latents is preserved in anyway across regions, right? Concretely, I am not sure I understand that reading through the columns of $C^m$ can tell us which dimensions are shared and which are not unless there is a post-processing step that reorders the latents (and the columns of $C^m$ respectively) so that the ordering of latents across regions is preserved (example: if latent 1 in regions 2 is correlated with a latent in region 1, then does it occupies the first dimension in the latent vector corresponding to region 1 at any time point?). Perhaps this falls our of the bayesian structure of the model, in which case it would be helpful to clarify this.
2. It would be nice to also include a literature review / related work section discussing the connections of mDLAG to multi-region LDS based models. Including this in the introduction is also fine, but I do think that it would be helpful to describe the pros and cons of both.
3. Finally, discussions of the amount of time and data needed to obtain reliable estimates from mDLAG would help practitioners understand the merits of the approach. Currently, there is a figure in the supplement describing the clock time per iteration, but as a reader I do not know how many iterations are needed in practice to fit the model, so having a number describing the overall runtime would be useful.

**Questions:**

In addition to the questions in the weakness section, I have some other minor questions:
1. Shouldn't $\bf{d}^m$ have a prior that requires it to be positive? I'd be curious to hear how the authors deal with this?
2. I am curious to understand the amount of data needed to reliably estimate the parameters of mDLAG, and so I wonder if the authors have any plots demonstrating accuracy vs # of data points for a given # of regions and population size.















**Limitations:**

I do not envision any societal implications of this work in the short term.

---

> ### Author Rebuttal · Authors · 2023-08-10
>
> > **Weaknesses**
>
> > 1. The main question that I found myself struggling with...
>
> We thank the reviewer for the opportunity to clarify this point. The ordering of latents across regions is preserved by definition of the model. Let us write out the observation model (Eq. 1) more expansively:
> 	$$y^m_{n,t} = \mathbf{c}^m_1 \cdot x^m_{n,1,t} + … + \mathbf{c}^m_p \cdot x^m_{n,p,t}  + \mathbf{d}^m + \boldsymbol{\varepsilon}^m$$
> From here, we can see that latent variable 1, $x^m_{n,1,t}$, always maps to population $m$'s neural activity through the 1st column of $C^m$ ($\mathbf{c}^m_1$), latent variable 2, $x^m_{n,2,t}$, always maps to population $m$'s neural activity through the 2nd column of $C^m$ ($\mathbf{c}^m_2$), and so on. This ordering of the latents and columns of $C^m$ is the same for every population, by definition. We have added the following clarification to Section 2:
> - "In particular, the $j$th column of $C^m$ maps the $j$th latent variable $x^m_{n,j,t}$ to population $m$."
>
> > 2. It would be nice to also include a literature review...
>
> We thank the reviewer for this suggestion. We have added the following text to the Discussion:
>
> "For exploratory data analysis, mDLAG's GP-based description of multi-population temporal structure is advantageous over an alternative linear dynamical system (LDS)-based description (Semedo et al., 2014; Glaser et al., 2020) in two respects: (1) a GP can be useful for exploratory data analyses where an appropriate parametric dynamical model is unknown *a priori*, and (2) mDLAG's continuous-time model enables the discovery of wide-ranging delays with high precision, which, in contrast to discrete-time LDS approaches, are not limited to be integer multiples of the sampling period or spike count bin width of the neural activity. Ultimately, these approaches can be complementary: one can use mDLAG to generate data-driven hypotheses about motifs of concurrent signaling across populations, and then test these hypotheses with a dynamical system-based approach."
>
> > 3. Finally, discussions of the amount of time and data needed...
>
> Regarding the amount of data needed, please see the General Response to Reviewers, "Amount of data needed."
>
> Regarding the amount of time needed, we have added the following text to Supplementary Fig. S3: "We conservatively ran each mDLAG model for 50,000 iterations, resulting in an average total runtime of 34 hours. Had we used a less conservative, but still reasonable, stopping tolerance of $10^{-6}$, the average number of iterations required for convergence would have been 17,000 (with similar parameter estimates), for an average total runtime of 11.5 hours." For further discussion, please see the General Response to Reviewers, "Computational demands."
>
> > **Questions**
>
> > 1. Shouldn't $\mathbf{d}^m$ have a prior that requires it to be positive?
>
> We thank the reviewer for this clarification question. Please see the General Response to Reviewers, "Positivity constraints on the mean parameter."
>
> > 2. I am curious to understand the amount of data needed...
>
> We thank the reviewer for the interesting question. Please see the General Response to Reviewers, "Amount of data needed."

---

> > ### Comment · Reviewer_qoeh · 2023-08-14
> > **Post rebuttal**
> >
> > Thank you for your detailed and thorough responses. I continue to think this is a worthwhile contribution to the NeurIPS community.

---

### Official Review · Reviewer_DL1P · 2023-07-04

**Soundness:** 3 good
**Presentation:** 4 excellent
**Contribution:** 3 good
**Rating:** 7
**Confidence:** 4

**Summary:**

The recent developments in neural recording technologies allow recording from large populations of neurons from multiple brain regions simultaneously. Latent space models are often used to analyze these datasets, but they are generally limited to the study of single or two populations of neurons. This work expands on existing dimensionality reduction methods and introduces a new probabilistic method to characterize interactions across multiple (more than two) populations of neurons. The proposed model can capture local and shared variability across multiple populations as well as the direction of the signal flow between areas and their temporal evolution with trial resolution. The authors validated the method on simulated data and neural data, showing that the model outperforms an existing method without temporal smoothing, GFA, and lesion versions of the model.

**Strengths:**

The paper is clearly presented and technically sound. The method was tested and shown to work well in simulation and neural data. This method builds on existing latent space models to enable multi-area signal characterization. Understanding inter-area population activity is an interesting problem in neuroscience, and this work proposes a new method that can help explain these emerging datasets. Importantly, even when tested in cases with only two neural populations, for which there are multiple existing methods, the proposed model still outperforms alternatives. This suggests that this tool could be broadly applied to single, two, or more than two areas population recordings. Additionally, when applied to neural data, they demonstrated the power of the approach by reporting new discoveries.

**Weaknesses:**

While the authors show the promise of their method to understand multi-area signals, they overlooked other existing methods that can also capture this information, such as multiset CCA or extensions of Procrustes alignment to multiple datasets. It would be interesting to add a performance comparison to these models. The model assumes linearity and Gaussian observations, but neural activity is often captured in spiking activity, which is better captured by a Poisson model. The authors show no indication of this limitation. Lastly, the computational cost of using a GP prior for temporal smoothing is relatively high compared to alternative methods, which could limit the potential applications. The method seems to be a direct extension of DLAG to multiple areas. If so, an explicit comparison between them could help assess the significance of the work.

**Questions:**

The simulated data has predetermined parameters, some set to match characteristics of the neural recordings and some others not. For example, the number of neurons in the recordings is in the order of dozens, while in one of the simulations, it is set to 10. What prompted the specific parameter choices? Or more interestingly, how robust is the model to potential changes in the different parameters: number of neurons, noise parameters, temporal delays, and so on? Another additional application listed for future work is the analysis of other signals like behavior. It would be relevant to know in these cases if the model requires similar dynamics or temporal binning for it to work.

**Limitations:**

The authors address the limitations of linearity and temporal smoothness but fail to address the potential limitations in computational cost and data demands.

---

> ### Author Rebuttal · Authors · 2023-08-10
>
> > **Weaknesses**
>
> > While the authors show the promise of their method to understand multi-area signals, they overlooked other existing methods...
>
> We thank the reviewer for pointing out these alternative methods. We agree that they are relevant, and we cite several review papers that mention such methods (Semedo et al., 2020; Kang et al., 2020; Keeley et al., 2020; Zhuang et al., 2020; Machado et al., 2022). In summary, these methods fall short in addressing two key challenges in the study of multi-population recordings (Fig. 1): (1) distinguishing network-level interactions, and (2) disentangling concurrent signal flow. Below and in Reviewer Figs. R2 and R3, we demonstrate these points empirically for multiset CCA. We believe that our empirical comparisons of mDLAG to group factor analysis (GFA; Fig. 4, Supplementary Fig. S2, Supplementary Fig. S3) are a representative demonstration of the advantages mDLAG has over the broad class of static methods (i.e., methods that do not consider the flow of time), which includes multiset CCA, Procrustes alignment, and GFA.
>
> In greater detail, we applied multiset CCA to both of our simulated datasets (Reviewer Fig. R2) and to the Neuropixels recordings (Reviewer Fig. R3):
> - When applied to Simulation 1 (Fig. 3), the loading matrix estimated via multiset CCA (Reviewer Fig. R2a, right) did not clearly show the group structure in the ground truth loading matrix (Reviewer Fig. R2a, left). Indeed, typical formulations of multiset CCA do not aim to distinguish these network-level interactions, unlike mDLAG or GFA.
> - When applied to Simulation 2 (Fig. 4, Reviewer Fig. R2b), multiset CCA's latent estimates represented a mixture of the two directed interactions (Reviewer Fig. R2c). GFA, a static method like multiset CCA, exhibited the same shortcoming (Fig. 4c). mDLAG successfully disentangled the two interactions (Fig. 4b).
> - When applied to the Neuropixels recordings (Fig. 5), multiset CCA was outperformed by both mDLAG (Reviewer Fig. R3a) and GFA (Reviewer Fig. R3b). These results are consistent with prior work in which GFA was shown to outperform multiset CCA on multi-area fMRI data (Klami et al., 2015), and DLAG was shown to outperform CCA on electrophysiological recordings from two brain areas (Gokcen et al., 2022).
>
> > The model assumes linearity and Gaussian observations...
>
> We thank the reviewer for making this point. Please see the General Response to Reviewers, "Gaussian observation model applied to spike counts."
>
> > Lastly, the computational cost...
>
> We thank the reviewer for making this point. Please see the General Response to Reviewers, "Computational demands."
>
> > The method seems to be a direct extension of DLAG...
>
> The reviewer is correct that mDLAG is an extension of DLAG to multiple (more than two) neuronal populations. We directly compare mDLAG to DLAG throughout the text:
> 1. In Section 2, under "mDLAG special cases," we write that "In the case of two populations (M = 2), mDLAG is equivalent to a Bayesian formulation of DLAG."
> 2. In Section 3, under "Validating mDLAG on recordings from V1 and V2," along with Supplementary Figures S1 and S2a, we make an empirical comparison between mDLAG and DLAG:
> "mDLAG outperformed DLAG across all datasets (Supplementary Fig. S2a, points above the diagonal), suggesting that ARD provides an improved method of model selection over the constrained grid search method used for DLAG, while also avoiding grid search’s computational drawbacks (see Supplementary Section S5)."
> 3. In Supplementary Section S5, we outline the heuristic model selection approach employed for DLAG (Gokcen et al., 2022), and in the Discussion, we note that "Scaling this type of approach to three or more populations (or external experimental variables) would be difficult. mDLAG is thus an advance toward scaling to large-scale multi-population recordings…"
>
> > **Questions**
>
> > What prompted the specific parameter choices?
>
> We thank the reviewer for this clarification question. Particularly in Simulation 1 (Fig. 3), we aimed to match all dataset characteristics to typical neural data. However, we chose to set 10 neurons per population to facilitate visual demonstration of the results (specifically, the loading matrices in Fig. 3a). The results do not meaningfully change if we set population sizes closer to those of the neural recordings. For example, if we re-run Simulation 1, but set 100 neurons per population instead of 10, then the accuracy of estimates (Fig. 3a,b) looks qualitatively similar. In fact, performance slightly improves: $R^2$ between ground truth and estimated latents is 0.957 (compared to 0.936 originally), and mean delay error is 0.58 ms (compared to 1.14 ms originally). We note that we further validated mDLAG on previously studied neural recordings (Section 3, "Validating mDLAG on recordings from V1 and V2) to test mDLAG in a truly realistic setting.
>
> Please see also the General Response to Reviewers, "Amount of data needed."
>
> > Another additional application...
>
> There is precedent for using latent time series models to study both neural activity and behavioral signals. For example, Kao et al., *Nature Communications*, 2015 developed a latent dynamical model to study the relationship between spike counts of a single motor area population and arm kinematics. They used a common set of latents (and consequently the same latent dynamical model) to describe neural activity and arm kinematics. The arm kinematics were sampled to the same time resolution as the spike count bins. Using mDLAG, we could extend this idea to multiple brain areas and arm kinematics.
>
> > **Limitations**
>
> We thank the reviewer for these points. Regarding computational cost, we note that we report mDLAG's runtime per EM iteration and compare to GFA's runtime in Supplementary Fig. S3. Please see also the General Response to Reviewers, "Computational demands." Regarding data demands, please see the General Response to Reviewers, "Amount of data needed."

---

> > ### Comment · Reviewer_DL1P · 2023-08-15
> >
> > I thank the authors for the detailed responses and their additional model comparisons. This is relevant work adding another tool to the existing methods for neuroscience research.

---

### Official Review · Reviewer_DK8X · 2023-07-06

**Soundness:** 3 good
**Presentation:** 3 good
**Contribution:** 3 good
**Rating:** 7
**Confidence:** 4

**Summary:**

This paper extends the delayed latents across groups (DLAG) to a recurrent general form (mDLAG), which allows analyzing the contribution of each latent dimension for multiple observational groups. Besides, the newly proposed mDLAG is able to identify the complicated directions of the information flow between groups with the corresponding time delay estimations. Experiments on two synthetic and two empirical datasets show the effectiveness of the mDLAG in analyzing multiple-grouped neural data.

**Strengths:**

* The model derivation is logically clear. The authors provide a very detailed clarification of the notation, parameter set, and generative procedure of this generative model
* The experiments are exhaustive, including two synthetic and two real-world datasets with very comprehensive analyses.
* The model is intuitive and in a general form. Its special cases can reduce the model to some famous generative model like the Gaussian process factor analysis (GPFA).

**Weaknesses:**

* Some math notation is a bit hard to understand, especially the $\boldsymbol x$. From my understanding, it is a high-dimensional tensor, but the author does not define this tensor in a clear way, e.g., the order of the dimensions.

**Questions:**

* It is hard for me to understand Fig. 1b.
* Eq. 3, theoretically mean firing rate $\boldsymbol d^m$ cannot have a Gaussian prior with zero means. A typical way is to model the log or logit of the firing rate since firing rates > 0.
* Is that $\boldsymbol x_{n,t}^m$ should be $\boldsymbol x_{n,t}$, since from Fig. 3b, it seems like you have a shared 7-d latent across all three neural populations A, B, and C?
* Is the inference very time-consuming?
* Is there any other models that can solve the same task with some tiny modification, so that the performance can be compared?

**Limitations:**

/

---

> ### Author Rebuttal · Authors · 2023-08-10
>
> >**Weaknesses**
>
> We realize we could be more explicit in defining our notation for the latents, $\mathbf{x}$. We have added the following text throughout Supplementary Section S1:
> - "we define latent variable $j$ (out of $p$) in population $m$ at time $t$ on trial $n$ as $x^m_{n,j,t} \in \mathbb{R}$."
> - "we represent the collection of all $p$ latent variables in population $m$ at time $t$ on trial $n$ as the vector $\mathbf{x}^m_{n,:,t} \in \mathbb{R}^p$."
> - "the latent variables in population $m$ on trial $n$ can be grouped into the matrix $X^m_n = [\mathbf{x}^m_{n,:,1} \cdots \mathbf{x}^m_{n,:,T}] \in \mathbb{R}^{p \times T}$. We represent a row of $X^m_n$ (i.e., the values of a single latent variable $j$ at all time points on trial $n$) as $\mathbf{x}^m_{n,j,:} \in \mathbb{R}^T$. Finally, we can form the three-dimensional array $X^m$ by concatenating the matrices $X^m_1,\ldots,X^m_N$ across trials along a third dimension."
>
> >**Questions**
>
> >- It is hard for me to understand Fig. 1b.
>
> Fig. 1b illustrates the geometric intuition behind mDLAG's ability to disentangle concurrent signal flow between a pair of populations. This task is difficult given measurements of raw neural activity, i.e., the activity along axes $A_1$ and $A_2$ in population A's neural state space (Fig. 1b, left), and the activity along axes $B_1$ and $B_2$ in population B's neural state space (Fig. 1b, right). However, if it were possible to measure activity along alternative dimensions in each population's state space (Fig. 1b, magenta and gray dimensions), then it could be possible to tease apart signals concurrently relayed in opposite directions. For example, if one were to measure activity along the magenta dimension in both population A and B (Fig. 1b, middle, top black time courses), then it is apparent that the activity in population A (top trace) leads the activity in population B (bottom trace). Similarly, if one were to measure activity along the gray dimension in both population A and B (Fig. 1b, middle, bottom black time courses), then it is apparent that the activity in population B (bottom trace) leads the activity in population A (top trace).
>
> To improve clarity, we have added the following to Fig. 1b: (1) labels for "Population A" (left) and "Population B" (right) to indicate which population corresponds to which set of axes, (2) explicit labels to indicate that axes $A_1$, $A_2$, $B_1$, and $B_2$ correspond to individual neurons, and (3) arrowheads on the gray and magenta dimensions to indicate the positive direction of the corresponding latent activity.
>
> If the reviewer has any other suggestions to help clarify the panel, please let us know.
>
> >- Eq. 3...
>
> We thank the reviewer for this clarification question. Please see the General Response to Reviewers, "Positivity constraints on the mean parameter."
>
> >- Is that $\mathbf{x}^m_{n,t}$...
>
> Thanks to the reviewer's comment, we realized that we could have been clearer in indicating how we chose to display the latent variables. $\mathbf{x}^m_{n,t}$ is still correct. For Fig. 3b, for example, the complete output would be 21 latent time courses (7 latents, time-delayed for each of the 3 populations). For concision, in Fig. 3b and throughout the paper, we display the set of latents for one of the populations. We have added the following text to the legends of Figs. 3 and 4:
> - "For concision, we show only latents corresponding to one population ($\mathbf{x}^m_{n,j,:}$); the remaining latents are time-shifted versions of those shown here."
>
> >- Is the inference...
>
> Across all datasets, the average runtime per iteration was 2.5 seconds (Supplementary Fig. S3). We have also added the following text to Supplementary Fig. S3: "We conservatively ran each mDLAG model for 50,000 iterations, resulting in an average total runtime of 34 hours. Had we used a less conservative, but still reasonable, stopping tolerance of $10^{-6}$, the average number of iterations required for convergence would have been 17,000, for an average total runtime of 11.5 hours." For further discussion, please see the General Response to Reviewers, "Computational demands."
>
> >- Is there any other models...
>
> We include the following qualitative and quantitative comparisons to alternative models throughout the text:
> - Section 3, Simulation 1, Fig. 3a: We compare an mDLAG model with automatic relevance determination (ARD) to an mDLAG model without ARD. We conclude that "The mDLAG model with ARD … recovered the population-wise sparsity structure with high accuracy (Fig. 3a, center). The mDLAG model without ARD, however, produced an estimate of the loading matrix with mostly non-zero elements (Fig. 3a, right)."
> - Section 3, Simulation 2, Fig. 4: We compare mDLAG to group factor analysis (GFA). We conclude that "mDLAG latent variable and time delay estimates accurately reflected the distinct signaling pathways across the three populations (Fig. 4b; $R^2$ between ground truth and estimated time courses: 0.989; mean delay error: 0.16 ms). Each latent estimated by GFA, however, notably reflected a mixture of both interactions (Fig. 4c, each latent time course exhibits two peaks)."
> - Section 3, V1-V2 recordings, Supplementary Figs. S1, S2a: We compare mDLAG to DLAG. We conclude that "mDLAG outperformed DLAG across all datasets (Supplementary Fig. S2a, points above the diagonal)..."
> - Section 4, Supplementary Figs. S2bc, S3: We compare mDLAG to GFA and a lesioned version of mDLAG in which time delays were fixed at zero ('mDLAG-0'). We conclude that "GFA, mDLAG-0, and mDLAG exhibited increasingly better leave-group-out prediction (Supplementary Fig. S2b: mDLAG-0 better than GFA; c: mDLAG better than mDLAG-0)..."
> - See also Reviewer Figs. R2 and R3, in which we show that mDLAG outperforms multiset CCA.
>
> To our knowledge, these are the most comparable methods to mDLAG. If the reviewer is aware of other related methods that we should cite or compare to, please let us know.

---

> > ### Comment · Reviewer_DK8X · 2023-08-16
> >
> > Thank you for your replies. Same as other reviewers, I also still think this is a good paper without major problems.

---

### Author Rebuttal · Authors · 2023-08-10

## General Response to Reviewers

We thank the reviewers for their constructive comments, which helped to strengthen our submission. Here, we provide responses to comments shared by multiple reviewers.

### Amount of data needed

Reviewers qoeh and DL1P inquired about the amount of data needed to fit an mDLAG model. In general, the answer to this question is highly data-dependent: the number of trials, number of neurons, number of populations, trial lengths, latent dimensionality, latent temporal structure, and noise levels all interact to influence mDLAG's performance. However, we can address this question with an empirical example (Reviewer Fig. R1). We re-fit mDLAG models to the Neuropixels dataset analyzed in Fig. 5, but we limited the number of experimental trials available in the training set (i.e., we fit mDLAG to training sets with 10, 25, 50, 100, 150, up to 225 trials—the full training set size). With as few as 100 training trials (less than half of the full training set size), mDLAG's test performance suffered by little more than 5% (Reviewer Fig. R1, black curve). Furthermore, with as few as 25 training trials, mDLAG still outperformed the group factor analysis (GFA) model fit to all 225 training trials (Reviewer Fig. R1, red dashed line).

This example demonstrates empirically that mDLAG performs well with trial counts typical of neurophysiological experiments. In principle, this data-efficiency is due to mDLAG's incorporation of (1) dimensionality reduction, (2) temporal smoothing, and (3) automatic relevance determination. These components act as forms of regularization that benefit model performance in data-limited regimes.

### Computational demands

Reviewers EMi6 and DL1P have pointed out that temporal smoothing via a GP prior is computationally demanding. We agree, and acknowledge in Supplementary Fig. S3 (where we compare mDLAG and GFA runtimes) that "Overall, mDLAG is more computationally intensive than GFA due to the incorporation of Gaussian processes. Each mDLAG fitting iteration requires the inversion of a $MpT \times MpT$ matrix (equation S11)."

We emphasize, however, two points from the Discussion: (1) mDLAG provides a tractable approach to model selection for neural recordings involving three or more populations, a problem that was previously computationally prohibitive; (2) The problem of scaling the latent inference step for this class of Gaussian process latent variable model has been studied extensively (Gallagher et al., 2017; Zhao et al., 2017; Duncker et al., 2018; Keeley et al., 2020; Jensen et al., 2021; Dowling et al., 2023). Approaches include diverse uses of Fourier approximations, inducing points, and additional variational approximations. mDLAG is compatible with these approaches, and we believe that they provide a promising avenue for continued computational development of the mDLAG framework.

### Gaussian observation model applied to spike counts

Reviewers EMi6 and DL1P have pointed out that mDLAG incorporates a Gaussian observation model, whereas the spike counts to which we apply mDLAG are generally better described by a Poisson observation model. While we agree with this point, we note that this limitation is not unique to mDLAG, and is shared by widely used methods such as factor analysis, Gaussian process factor analysis, and latent linear dynamical systems. Intuitively, one can think of the Gaussian model as capturing the first and second moment of the spike count data, rather than requiring that the data be Gaussian-distributed. In our experience, for neural recordings similar to those analyzed in this work, we have not encountered a scientific finding that could be seen with a Poisson or point process observation model that could not also be seen using a Gaussian model. Hence mDLAG is still widely applicable to spiking neural activity. We also emphasize, as we have in the Discussion, that mDLAG's Gaussian observation model enables wider applicability to other neural recording modalities. For spiking neural activity in which a Gaussian observation model is meaningfully limiting, a Poisson or point process observation model can be successfully incorporated, but at the expense of increased computational requirements (Zhao et al., 2017; Duncker et al., 2018; Keeley et al., 2020; Jensen et al., 2021; Dowling et al., 2023).

### Positivity constraints on the mean parameter

Reviewers qoeh and DK8X have inquired about potential positivity constraints on the mean parameter $\mathbf{d}^m$. We thank the reviewers for this question, as we realized we needed to further clarify our data preprocessing steps. Introducing a positivity constraint on $\mathbf{d}^m$ could make sense for applications to neural spike counts, especially if it is particularly important to interpret each element of $\mathbf{d}^m$ as the mean firing rate of each neuron across time and trials. However, a positivity constraint might not make sense for analyses of different recording modalities, or given certain preprocessing choices of spiking neural activity. For example, in this work, because we were interested in inter-population interactions on timescales within a trial, we subtracted the mean across time bins within each trial from each neuron. This step removed activity that fluctuated on slow timescales from one stimulus presentation to the next (Cowley et al., 2020). Consequently, the data input to mDLAG were no longer positive-valued spike counts. We have added the following text:
- Section 3: "Because we were interested in V1-V2 interactions on timescales within a trial, we subtracted the mean across time bins within each trial from each neuron. This step removed activity that fluctuated on slow timescales from one stimulus presentation to the next (Cowley et al., 2020)."
- Section 4: "As we did for the V1-V2 recordings, above, we subtracted the mean across time bins within each trial from each neuron, to remove slow fluctuations beyond the timescale of a trial."

---

### Decision · Program_Chairs · 2023-09-21

**Decision:**

Accept (spotlight)

**Comment:**

This work provides a computational model for identifying the network interactions between multiple brain areas. The work builds off of a 2-area model, delayed latents across groups (DLAG), which finds low-rank representations with temporal delays between brain areas, representing the expected propagation time across neural populations. The model establishes a Bayesian fitting procedure that generalizes on the past DLAG method, using Gaussian Process latent states and ARD determination of neural subsets corresponding to each state. The method is shown on simulated data as well as real visual cortex data across three neural populations recorded with SOTA neuropixel probes.

The reviewers universally found this work interesting and timely with the increased capability to record from multiple brain areas simultaneously. The main remaining weakness is one of scaling (how will the method scale in computation and interpretation to 40+ brain areas?). Regardless this is a strong paper that I recommend for acceptance to NeurIPS.